# N-Terminal Acetyltransferase Naa40p Whereabouts Put into N-Terminal Proteoform Perspective

**DOI:** 10.3390/ijms22073690

**Published:** 2021-04-01

**Authors:** Veronique Jonckheere, Petra Van Damme

**Affiliations:** Department of Biochemistry and Microbiology, Ghent University, K. L. Ledeganckstraat 35, 9000 Ghent, Belgium; Veronique.Jonckheere@ugent.be

**Keywords:** N-terminal proteoforms, N-alpha acetyltransferase, N-terminal acetylation, NAA40, Naa40^S^, biotin-dependent proximity labeling (BioID), alternative translation initiation, N-terminal myristoylation, NMT1

## Abstract

The evolutionary conserved N-alpha acetyltransferase Naa40p is among the most selective N-terminal acetyltransferases (NATs) identified to date. Here we identified a conserved N-terminally truncated Naa40p proteoform named Naa40p25 or short Naa40p (Naa40^S^). Intriguingly, although upon ectopic expression in yeast, both Naa40p proteoforms were capable of restoring N-terminal acetylation of the characterized yeast histone H2A Naa40p substrate, the Naa40p histone H4 substrate remained N-terminally free in human haploid cells specifically deleted for canonical Naa40p27 or 237 amino acid long Naa40p (Naa40^L^), but expressing Naa40^S^. Interestingly, human Naa40^L^ and Naa40^S^ displayed differential expression and subcellular localization patterns by exhibiting a principal nuclear and cytoplasmic localization, respectively. Furthermore, Naa40^L^ was shown to be N-terminally myristoylated and to interact with N-myristoyltransferase 1 (NMT1), implicating NMT1 in steering Naa40^L^ nuclear import. Differential interactomics data obtained by biotin-dependent proximity labeling (BioID) further hints to context-dependent roles of Naa40p proteoforms. More specifically, with Naa40^S^ representing the main co-translationally acting actor, the interactome of Naa40^L^ was enriched for nucleolar proteins implicated in ribosome biogenesis and the assembly of ribonucleoprotein particles, overall indicating a proteoform-specific segregation of previously reported Naa40p activities. Finally, the yeast histone variant H2A.Z and the transcriptionally regulatory protein Lge1 were identified as novel Naa40p substrates, expanding the restricted substrate repertoire of Naa40p with two additional members and further confirming Lge1 as being the first redundant yNatA and yNatD substrate identified to date.

## 1. Introduction

Largely believed to occur co-translationally when a protein N-terminus emerges from the ribosomal peptide exit channel, N-terminal acetylation (Nt-acetylation) represents a widespread modification affecting most intracellular proteins in higher eukaryotes [1]. Because of its diverse functional outcomes, however, its consequences remain largely underexplored. Although the responsible enzymes—N-terminal acetyltransferases or NATs—and their substrate specificities are generally conserved in eukaryotes (reviewed in [2]), a higher eukaryotic-specific NAT, namely NatF [3] was recently found to partially account for the observed variations in Nt-acetylation patterns among different eukaryotes [1,4,5]. Furthermore, while representing an infrequent modification in prokaryotes [6,7,8], post-translational Nt-acetylation has also been observed in archaea and in plant [9], and more recently, a dedicated NAT responsible for post-translational Nt-acetylation of actin was discovered [10].

To bring the catalytic NAT subunit—responsible for the transfer of the acetyl moiety from acetyl-coenzyme A to the α-amino group of the N-terminal amino acid—in close proximity to the nascent polypeptide, NATs frequently comprise one or more auxiliary subunit(s) implicated in ribosome binding [2]. However, N-alpha acetyltransferase 40 (Naa40p) also known as NatD, Nat11, Patt1 or Nat4, was shown to be active as a monomeric enzyme. Furthermore, as compared to most other NATs displaying relative extensive substrate repertoires and with NatA putatively targeting up to ~50% of the proteome [3], Naa40p displays a restricted substrate specificity over a few specific histone N-termini and therefore represents one of the most substrate-selective NATs identified to date [11]. More specifically, Naa40p activity over histones H2A and H4 was originally discovered by identification of the Nt-acetylated Ser-starting N-termini of H2A and H4 in yeast deleted for NatA (yNatAΔ), a type of N-termini typically targeted by NatA [3]. Both yeast (*NAT4*, subsequently referred to by the alias y*NAA40* [12]) [13] and human *NAA40* (h*NAA40*) [14] were previously cloned and characterized, and we reported that these orthologs genes encode proteins displaying overlapping substrate specificities [14]. Interestingly, while ribosome-association was found for both yeast and human orthologs, hNaa40p and mouse Naa40 (mNaa40p) was reported to additionally locate in the nucleus [14,15], indicating that Naa40p might carry out co- as well as post-translational activities [14,16].

Due to the additional presence of a unique and complementary substrate binding pocket extending the specificity beyond the second amino acid residue of cognate substrates, Naa40p is exceptional in that it has sequence requirements for the first four residues of its substrates [11,14]. More specifically, X-ray structure determination of Naa40p revealed structural features tailored for specifically accommodating the N-termini of H4 and H2A [11] (i.e., Ser-Gly-Arg-Gly- starting N-termini in case of human H4 and H2A).

Interestingly, while displaying an overall mixed α/β GCN5-like fold similar to other NAT structures, Naa40p possesses a unique N-terminal segment adopting an extra helix-loop-strand secondary structural element (residues 24–51 in hNaa40p) which is implicated in cognate substrate binding and stability maintenance of the catalytic core of Naa40p, thereby serving a potential analog role as compared to the auxiliary subunits of other major NAT complexes [11].

Notably, *S. cerevisiae* H2A has a divergent Ser-Gly-Gly-Lys- N-terminal sequence and hNaa40p was unable to restore Nt-acetylation of yeast H2A (yH2A) in contrast to being able to restore yH4 Nt-acetylation, indicating that yNaa40p likely evolved to specifically accommodate both divergent yeast histone N-termini.

Besides histone H4 and H2A being confirmed Naa40p substrates, we previously reported four other putative yNaa40p substrates. More specifically in case of Scl1, Ypi1, Dph1 and Lge1, their corresponding Ser-Gly- starting N-termini remained (partially) Nt-acetylated in yeast deleted for yNatA [17], an observation in line with H2A and H4 being Nt-acetylated in the same yeast background.

Although deletion of *NAA40* in yeast (y*naa40*Δ) was associated with reduced growth in the presence of various chemicals such as transcription inhibitors—a phenotype which was enhanced upon additional mutation of H4 lysine residues targeted for protein modifications making up the histone code [16]—it was only in 2013 that a first molecular and biological function of H4 Nt-acetylation was elucidated. More specifically, by the observed inverse correlation and interplay between H4 Nt-acetylation and asymmetric Arg3 methylation in regulating ribosomal RNA (rRNA) gene expression, Schiza et al. proclaimed H4 Nt- acetylation to serve as a sensor for cell growth [18]. Consistently, h*NAA40* knockdown in colon cancer cells was shown to reduce rRNA expression [19], hinting to a similar role for both orthologs in regulating rDNA expression and consequently ribosome biogenesis. Furthermore, by its observed aberrant expression in hepatocellular carcinoma and high variable expression in different cancer types (Human Protein Atlas project), Naa40p has been implicated in disease [15] and was additionally shown to play a role in hepatic lipid metabolism [20].

In the present study, we identified a conserved 21 amino acid Nt-truncated Naa40p proteoform named Naa40p25 or short Naa40 (Naa40^S^). As compared to canonical Naa40p27 or long Naa40 (Naa40^L^), enzymatically active Naa40^S^ displayed a differential expression and subcellular localization pattern. Furthermore, by the observed association of hNaa40^S^ with the translational machinery and hNaa40^L^ being implicated in nucleolar ribosome biogenesis and ribonucleoprotein assembly, interactomics data untangled previously reported Naa40p activities. Overall, our data point to a possible proteoform-specific segregation of post- and cotranslational Naa40p proteoform activities.

## 2. Results

### 2.1. Identification of an Alternative N-Terminally Truncated Proteoform of Human N-Terminal Acetyltransferase Naa40

We previously reported that human h*NAA40* encodes a 237-amino acid (AA) long protein with strong similarity to yNaa40p [14]. As inferred from their overlapping substrate specificities, y*NAA40* and h*NAA40* were catalogued as orthologs genes [14]. In another study, studying the translation initiation landscape of mice and men and the expression of multiple molecular forms of proteins from a single gene (i.e., proteoforms), we reported the identification of an alternative, shorter hNaa40p proteoform by the identification of its corresponding Nt-acetylated N-terminus MDAVCAKVDAANR (Figure 1A), hereafter referred to as short Naa40 (Naa40^S^). Naa40^S^ is generated upon translation initiation at the second in-frame ATG codon downstream of the canonical, database-annotated translation initiation start site (dbTIS) ([21] and Figure 1A).

When looking at endogenous Naa40p expression by means of Western analysis, and while the canonical Naa40p proteoform of Naa40—Naa40p27 or long Naa40^L^—appears the most common proteoform of expressed h*NAA40* in human cells, expression of hNaa40^S^ could be demonstrated in certain cell lines such as the human (pro-)myelocytic leukemia cell lines HL-60 (Figure 1B) and K-562 (Figure 2B). Furthermore, the corresponding Nt-peptide of hNaa40^S^ was previously also identified by N-terminal proteomics analysis performed in Jurkat, HCT 116 and B-cells [21].

Interestingly, when differentiating K-562 cells with the phorbol ester and diacylglycerol analog phorbol 12-myristate 13-acetate (PMA) into megakaryocytes via the mitogen-activated protein kinase pathway [23] (Figure 2A), hNaa40^L^ expression remained stable over the differentiation process, whereas hNaa40^S^ was significantly down-regulated (Figure 2B,C), indicating that Naa40p proteoform expression can be differentially regulated.

Although the differentially spliced ENST00000542163.1 transcript can give rise to expression of the shorter 216 AA long hNaa40p proteoform hNaa40^S^ (Figure 3A), in vitro (and cellular, see below) transcription/translation assays making use of full-length h*NAA40* cDNA, indicated that hNaa40^S^ translation can also potentially arise from alternative translation initiation. When considering the Kozak motif (i.e., the consensus Kozak motif being gcc[A/G]ccAUGG), the likelihood of the second in-frame AUG codon to serve as downstream TIS (dTIS) was predicted 0.33 versus 0.92 for the first AUG codon (dbTIS) by translation initiation prediction at http://atgpr.dbcls.jp, accessed on 1 December 2020, overall indicating that the second in-frame AUG codon may potentially serve as an alternative TIS. Alternative TIS codon and sequence context alignment in the corresponding *NAA40* loci of representative mammalian genomic sequences (alignment set hg38_58 including human, chimp, mouse, rabbit, dog and cow genomic sequences) indicated that the alternative TIS and its surrounding Kozak consensus motif is highly conserved (Figure 3B). Further of note, this alternative TIS also appears conserved in *NAA40* sequences of more distantly related organisms such as *Xenopus tropicalis* (B1H2T3), *Gallus* (A0A1D5P6P4) and *Danio rerio* (Q568K5).

### 2.2. hNaa40^S^ N-Terminally Acetylates Yeast Histone H4

Since Naa40^S^ retains the GNAT (GCN5-related N-acetyltransferase superfamily) fold (aa 63-216 in hNaa40^L^), a domain implicated in Ac-CoA binding and substrate recognition [28], we assessed whether hNaa40^S^ was enzymatically active and displayed the same substrate specificity and physiological substrate repertoire as hNaa40^L^. For this, we ectopically expressed hNaa40^L^ and hNaa40^S^ in yeast as we previously demonstrated that hNaa40 expression rescued H4 Nt-acetylation in yeast lacking *NAA40* (y*naa40*Δ). Specific expression of the hNaa40^S^ and hNaa40^L^ proteoforms in yeast was confirmed by Western blot analysis (Figure 4). Histone extractions followed by in-gel-stable isotope labeling (ISIL), protein digestion and LC-MS/MS was performed to assess the *N*-acetylation status of histone N-termini as described previously [14]. When comparatively analyzing the *N*-acetylation states of identified histone N-termini originating from control y*NAA40*, y*naa40*Δ and h*NAA40* or h*NAA40^S^* yeast, we found that the previously characterized yNaa40p and hNaa40^L^ substrate histone H4 was also Nt-acetylated when expressing hNaa40^S^, while as expected, the *N*-free form of the Ser-Gly-Gly-Lys- starting N-terminus of yH2A—previously identified as being an exclusive yNaa40p substrate—could only be observed in the hNaa40^S^ expressing strain ([14] and data not shown). Furthermore, as inferred from the *N*-acetylation states in the different yeast setups analyzed, the conserved and replication independent yeast H2A variant H2A.Z, encoded by *HTZ1* (*YOLO12C*) and differing in its N-terminal sequence from the other two yeast H2A isoforms (H2A.1 and H2A.2), was here identified as a novel Naa40p yeast substrate. More specifically, the Ser-Gly-Lys-Ala- starting N-terminus of H2A.Z was identified as being Nt-acetylated in the control (y*NAA40*) and both h*NAA40* setups, while the *N*-free form was only observed in the y*naa40*Δ setup (Figure 4). Overall, the discovery of H2A.Z as a novel Naa40p substrate adds yet another histone member to the limited set of physiological Naa40p substrates identified to date.

### 2.3. The Transcriptionally Regulatory Protein Lge1 Represents a Redundant yNatA/yNatD Substrate

In pursuit of identifying novel Naa40p substrates besides histones H2A, H4 and H2A.Z, we comparatively analyzed the Nt-acetylomes of control (y*NAA40*) and y*naa40*Δ yeast by means of N-terminal proteomics [25,29,30]. As inferred from the significantly reduced degree of Nt-acetylation of the N-terminal peptide (SGYTGNNYSR) of the yeast transcriptional regulatory protein Lge1 in the y*naa40*Δ setup (i.e., 100% Nt-acetylated in y*NAA40* yeast opposed to only partially Nt-acetylated (73%) in y*naa40*Δ yeast, respectively), Lge1 was identified as a novel yNaa40p substrate. Since in our previous study reporting on NatA substrates [17], Lge1 also represented one of the four Ser-Gly-starting N-termini which displayed a significantly reduced degree—but not complete lack of—Nt-acetylation upon yNatA deletion (82% Nt-acetylated in yNatA-Δ [17]), combined this data confirms that yNatA as well as yNaa40p display redundant activity over Lge1 thereby identifying Lge1 as being the first redundant yNatA/yNaa40p substrate.

### 2.4. hNaa40^S^ Expression in hNAA40^L^ Knockout Human Haploid Cells Is Accompanied with N-Terminally Free Human Histone H4 Expression

By producing a frameshift mutation in h*NAA40*, a selective h*NAA40^L^* knockout was created in the human haploid cell line HAP1 using CRISPR/Cas9 editing [31]. More specifically, exon 2 (NM_024771) was selected for guide RNA design with guide RNA sequence GAAGAAGCAGAAGCGGTTGG and a HAP1 clone containing a 4 bp. deletion in exon 2 was obtained as confirmed by Sanger and RNA-sequencing (data not shown). Expression of both hNaa40p proteoforms could be detected in HAP1 control cells but with hNaa40^S^ displaying a much lower expression as compared to hNaa40^L^ (Figure 5A). Since in theory, the frameshift still permits hNaa40^S^ proteoform expression, Western blot analysis probing for endogenous hNaa40p indeed confirmed the exclusive expression of hNaa40^S^ in two independent h*NAA40^L^* knockout (h*NAA40^L-^*) HAP1 clones (Figure 5A).

To assess if (limited) hNaa40^S^ expression was sufficient for maintaining hNaa40p substrate Nt-acetylation, we interrogated the *N*-acetylation status of the previously characterized hNaa40 substrate H4. Therefore, histone isolation followed by MS analysis was performed as described above. Intriguingly, in the h*NAA40^L-^* HAP1 cell line expressing residual hNaa40^S^, the complete *N*-free form of the Ser-Gly-Arg-Gly- H4 N-terminus could be observed, indicating that H4 remained (at least partially) Nt-free (Figure 5B). This observation may indicate that hNaa40p proteoforms may display non-redundant activities, and that expression of hNaa40^L^ is required for (efficient) Nt-acetylation of previously characterized histone substrates.

### 2.5. hNaa40^L^ and hNaa40^S^ Proteoforms Display a Differential Subcellular Localization

As such, despite the observed enzymatic activity of hNaa40^S^ over endogenous yNaa40p substrates, exclusive expression of hNaa40^S^ in HAP-1 cells was unable to rescue human H4 Nt-acetylation. Since it is inconclusive whether ectopic expression of hNaa40^S^ in yeast acts co- and/or post-translationally and since previously, a dual nuclear and cytosolic localization pattern of hNaa40p and mNaa40p could be observed [14,15], we examined the subcellular localization of both hNaa40p proteoforms in various cell lines (i.e., A-431, HEK 293T cells). For this purpose, various C-terminally Flag-tagged hNaa40p expressing constructs were generated, encoding either wild type full-length and thus also potential short hNaa40p (hNaa40^L/S^), hNaa40p expressing constructs with mutation of the dTIS (hNaa40^L(dTIS>CTG)^) or dbTIS (hNaa40^S(dbTIS>CTG)^) (i.e., mutation of the corresponding ATG codon—encoding either M22 or M1 of hNaa40p and serving as dTIS and dbTIS, respectively—to CTG), besides a construct exclusively encoding the corresponding hNaa40^S^ coding sequence (CDS). As expected, expression of these constructs either resulted in the expression of both hNaa40p proteoforms (hNaa40^L/S^), or solely the long (hNaa40 ^L(dTIS>CTG)^) or short (hNaa40 ^S(dbTIS>CTG)^ and hNaa40^S^) hNaa40p proteoform(s). Expression of hNaa40^S^ from the hNaa40^L/S^ and dbTIS mutated hNaa40p (hNaa40 ^S(dbTIS>CTG)^) constructs again indicates that alternative translation—and thus not only alternative splicing—can potentially give rise to hNaa40^S^ expression.

Interestingly, upon expression of the different constructs and performing cytoplasmic and nuclear-associated compartment fractionation using the non-ionic detergent NP-40, a clear enrichment of the hNaa40^L^ in the nuclear-enriched fraction (N) could be observed while hNaa40^S^ displayed a (predominant) cytoplasmic localization (C) (Figure 6), an observation which could be confirmed in at least 3 independent experiments and in various cell lines (data not shown). The dual localization pattern of hNaa40p previously observed [14] can thus (at least in part) be explained by the differential subcellular localization of the two hNaa40p proteoforms expressed.

### 2.6. Nuclear Localized hNaa40^L^ Harbors a Nucleolar Localization Sequence and Is N-Terminally Myristoylated

In line with the above observation of hNaa40^L^ being localized in the nucleus, the 23 aa long N-terminal sequence of hNaa40^L^ (MGRKSSKAKEKKQKRLEERAAMD) was predicted to hold a nucleolar localization sequence (NoLS) [32]. Although still ill-defined, NoLS and nucleolar retention signal (NoRS) sequences are rich in Arg and Lys residues and frequently overlap with nuclear localization signals (NLSs).

Furthermore, hNaa40^L^ was predicted as being Nt-myristoylated by the neural network predictor Myristoylator with very high confidence (score 0.99) [33] and also identified as being myristoylated using a proteome profiling method probing for substrates responsive to *N*-myristoyltransferase (NMT) inhibition [34]. More specifically, initiator Met (iMet)-processed protein sequences starting with an N-terminal glycine residue (e.g., Naa40^L^) may be co- or post-translationally modified by 14-carbon fatty acid addition through the action of NMTs (i.e., Nt-myristoylation). Since Nt-myristoylation has been implicated in regulation of protein localization and nuclear protein transport [35,36] among other regulations (as recently reviewed in [37]), we assessed the Nt-myristoylation status of transiently expressed hNaa40p.

For this, myrisate 12-azidododecanoic acid (azidomyristate)—a non-toxic alkyl azide analog of myristate previously shown to be converted to azidomyristoyl-CoA by fatty acyl CoA synthetase [38]—was used for the metabolic labeling of endogenous NMT protein substrates. As inferred from Western blot analysis following labeling of the alkyne through copper-catalyzed azide-alkyne cycloaddition (CuAAC) with alkyn-biotin conjugate, optimal metabolic protein incorporation was achieved after supplementing HCT 116 cell cultures for 8 (Figure 7) or 24 h (data not shown) with 50 to 100 μM azidomyristate.

By exploiting the biotin moiety of the alkyne-functionalized biotin used, selective enrichment of labeled proteins was performed. Although the negative control protein Tubulin-α (TUBA) remained unlabeled as demonstrated from the lack of TUBA signal in the respective eluates (lanes 5 in 50 and 100 µM Myr setups in Figure 7), the well characterized NMT substrates, being it the catalytic subunit of protein kinase A (PKA) PRKACA (cAMP-dependent protein kinase catalytic subunit alpha) [39] and the proto-oncogene tyrosine-protein kinase Src (i.e., cellular Src or c-Src) [35] were enriched by streptavidin-aided affinity purification in the azidomyristate treated samples (Figure 7). As such, we assessed the myristoylation status of transiently expressed hNaa40p and found that while hNaa40^S^ remained unlabeled (i.e., hNaa40^L^ was exclusively detected in the respective eluates), hNaa40^L^ was identified as being *N*-myristoylated and thus univocally identified as being an NMT substrate (Figure 7).

As in humans, myristoylation is catalyzed by NMT1 and NMT2 [40], it is noteworthy that NMT1 was identified as a hNaa40^L^ specific interaction partner using biotin-dependent proximity labeling (BioID) (see below). Combined, these results indicate that hNaa40^L^ represents an NMT1 substrate. Furthermore, since the N-terminus of NMT1 is thought to mediate ribosome interactions [40] we hypothesize that Nt-myristoylation of hNaa40^L^ by NMT1 occurs cotranslational.

### 2.7. Interactomics Profiling of hNaa40p Proteoforms by Proximity-Dependent Biotin Identification (BioID)

To study the subcellular localization of hNaa40p proteoforms in more detail and to unravel some of the (differential) proteins they interact with and consequently infer different hNaa40p proteoform functionalities, we applied proximity-dependent biotin identification or BioID (Figure 8).

For this purpose, a Flag-tagged version of the promiscuous biotin ligase protein BirA* was cloned in frame to the C-terminus of hNaa40^L(dTIS>CTC)^ (permitting only hNaa40^L^ expression) or hNaa40^S^ in a doxycycline-inducible expression vector. Since it has previously been reported that the eGFP-BirA* fusion protein localizes to both the nucleus and cytoplasm [41], eGFP-BirA* was used as an appropriate reference for promiscuous and non-specific protein labeling by BirA* in both the nuclear and cytosolic compartments (Figure 8A). After recombination-assisted stable integration into Flp-In T-REx 293 cells, stable inducible cell lines were created and doxycycline-induced expression and (auto-)biotinylation of BirA* fused proteoforms and eGFP verified by Western blot analysis (Figure 8). For biotin labeling of proximal proteins, cell cultures were supplemented with biotin for 24h. To limit overexpression effects and for comparable bait expression, optimized (low) doxycycline levels were chosen (experimental procedures and data not shown) (Figure 8B). Following labeling, biotinylated proteins were isolated using streptavidin agarose purification (Figure 8A). Silver staining of the streptavidin bound fraction, indicated a change in the specificity of biotinylation between eGFP-BirA* and the hNaa40p-BirA* setups (Figure 8C). Streptavidin bound proteins were subsequently analyzed by quantitative mass spectrometry (MS) (experimental procedures). To enable relative interactome comparisons, label-free quantification was performed by use of the intensity values (LFQ) obtained using the MaxLFQ algorithm [42]. After log2 transformation of the obtained LFQ values, correlation (Pearson) and principal component analysis (PCA) indicated high correlation and clustering among replicate samples while indicating the variability between the different BioID setups analyzed (Figure 9). A multiple analysis of variance (ANOVA) test was performed to reveal (differential) hNaa40p proteoform interactors (*p* ≤ 0.01). The intensities of differential interactors are shown as heat maps after z-score transformation and non-supervised hierarchical row clustering (Figure 10A). Interestingly, of the previously identified binary hNaa40p interactors [43] (i.e., the hNaa40p substrates; histone H4 and macro-H2A.1, the basic transcription factor BTF3 and its 62% identical BTF3 paralog BTF3L4), all 4 interactors were significantly enriched with a threshold *p* value of 0.01 in the hNaa40^L(dTIS>CTC)^ setup, indicative of their (specific) interaction with hNaa40^L^ (Appendix A and Figure 10A). As mentioned above, NMT1 was over 10-fold enriched in the hNaa40^L(dTIS>CTC)^ versus the Naa40^S^ setup, indicating that NMT1 is the likely N-myristoyltransferase of hNaa40^L^.

When specifically comparing the hNaa40^L(dTIS>CTC)^ and Naa40^S^ setups for the relative enrichment of gene annotations (UniProt keywords, GOCC (Gene Ontology Cellular Component), GOBP (Gene Ontology Biological Process) and GOMF (Gene Ontology Molecular Function) names, and CORUM complexes), 1D annotation enrichment analysis [44] pointed to the respective enrichment of nuclear and cytoplasmic proteins in the hNaa40^L(dTIS>CTC)^ and Naa40^S^ setups, respectively, matching the differential subcellular localizations of the two hNaa40p proteoforms observed. Furthermore, while hNaa40^L(dTIS>CTC)^ enriched annotations *p* ≤ 0.02) encompassed ‘ribosome biogenesis’, ‘ribonucleoprotein binding’, ‘RNA-dependent ATPase activity’ and ‘nucleolus’, hNaa40^S^ interactors were among others significantly enriched (*p* ≤ 0.02) for ribosomal protein, redox-active center containing proteins and proteins implicated in translation or fatty acid biosynthesis (Appendix A and Figure 10B). As such, proximal proteins and thus putative interactors of the long and short proteoform of hNaa40p match dedicated subcellular and thus segregated molecular functions such as ribosome biogenesis, translation and fatty acid synthesis where hNaa40p has all previously been shown to play a role in.

## 3. Discussion

The expression of multiple molecular forms of proteins (proteoforms) from a single gene, particularly driven by alternative translation initiation, has only recently been acknowledged to serve as a critical contributor in shaping translational landscapes across all kingdoms of life [21,45]. Although our original study on N-terminal proteoforms reported over 1500 N-termini pointing to translation initiation at dTIS, this source of protein diversity remains vastly uncharted. Data from the few functionally characterized proteoforms however suggest that alternative Nt-proteoforms may very likely display differential functions besides altered (sub-)cellular localizations or stability [45,46].

With this study, we set out to functionally characterize the newly identified shorter proteoform of the human N-alpha acetyltransferase Naa40p, hNaa40^S^, explaining the previously observed lower molecular 25 kDa weight band when probing for endogenous Naa40p [14]. Viewing diverging N-terminal Naa40p sequences, the presence of possible *NAA40* dTIS in other species such as *Saccharomyces cerevisiae*, *Drosophila melanogaster* and plant is less clear. Interestingly, in case of *Arabidopsis thaliana* Naa40p (At1g18335), N-terminal peptide evidence (Ac-MDLKKR) was found in the N-terdb database (n-terdb.i2bc.paris-saclay.fr) matching the expression of an Nt-acetylated AtNaa40p^S^ proteoform starting at AA position 28.

Interestingly, while both the canonical and short hNaa40p proteoforms exhibit indistinguishable enzymatic activity, hNaa40^L^ and hNaa40^S^ display a differential subcellular localization. Although the Human Protein Atlas originally reported that hNaa40p resides in nuclear bodies in case of A-431 and U2 OS cells, hNaa40p localization was more recently updated to nucleoplasm, with additional localizations to the cytosol and the centriolar satellite (https://www.proteinatlas.org/ENSG00000110583-NAA40, accessed on 1 December 2020). In our study, we observed that the dual subcellular localization pattern of Naa40p previously reported in mice and human can (in part) be explained by the differential subcellular localization of two Naa40 proteoforms expressed.

Interestingly, and in analogy, besides the previous report on differentially localized hNaa30p isoforms raised upon alternative splicing [47], a differential subcellular localization and stability was previously also reported in the specific case of mouse Naa10p isoforms differing at their C-terminus and with mNaa10p^S^ being an alternative splicing isoform specific to mouse found to be differentially regulated [48].

Moreover, we demonstrated that the N-terminal extension of hNaa40p^L^—which harbors a putative nucleolar localization sequence (NoLS) and which can be Nt-myristoylated—likely explains the predominant nuclear/nucleolar localization of hNaa40^L^. Interestingly, the N-terminus of AtNaa40p^L^ holds a ChloroP predicted chloroplast transit sequence (aa1-26) [49] and the subcellular localization database for Arabidopsis proteins (SUBA4) also predicts a nuclear localization of AtNaa40p [50], indicating that AtNaa40p proteoforms of which note above likely display differential subcellular localizations.

Furthermore, the unique extra helix-loop-strand secondary structural element contained within the N-terminal segment of hNaa40p implicated in substrate binding and stability maintenance, is contained within both hNaa40 proteoforms as it comprises residues 24–51 of full-length hNaa40p [11]. However it is important to note that the first 16 and last 17 amino acid residues of full-length hNaa40p which fall outside the ~150 residue core of high sequence conservation but which are respectively highly enriched in positively charged Lys/Arg, and His residues, were not present in the recombinant hNaa40 variant (i.e., hNaa40, residues 17–220) used for structural determination [11]. Therefore, if and how these terminal sequences may affect stability, substrate accessibility or binding is currently not known. Since we observed that C-terminal His-tagging of Naa40p resulted in a somewhat less outspoken subcellular segregation of hNaa40p proteoform localization (data not shown) it is conceivable that the C-terminal sequence might also impact hNaa40p localization.

Besides the proteoform-specific subcellular localization of hNaa40p proteoforms, differential expression of these proteoforms upon cellular differentiation could also be demonstrated. Viewing the aberrant and highly variable *NAA40* expression observed in a variety of cancers [15], expression profiling at the level of proteoforms thus appears a prerequisite to further unravel the (distinct) roles of Naa40p proteoforms in disease.

Furthermore, with only histone H4 and H2A proteins being reported as Naa40p substrates, we here expanded on the Naa40p substrate repertoire by the identification of yH2A.Z (SGKA) and the transcriptional regulatory protein Lge1 (SGYT) as endogenous yNaa40p and redundant yNaa40p/yNatA substrates, respectively. Furthermore, since we previously observed that only the Nt-acetylated counterpart of a synthetic histone H4 Nt-peptide was refractory towards hNaa40p mediated *N*-acetylation [14], hNaa40p acetylates the α-amino group of H4 and (not significantly) any of the lysine residues known to be Nε-acetylated by lysine acetyltransferases. Moreover, structural data revealed the striking divergent feature of Naa40p being the presence of occluding loops preventing internal lysines from inserting into the active site, providing a molecular basis for specific amino-terminal acetylation by Naa40p [11]. Altogether, the lack of *N*-free H4 and H2A.Z can thus be attributed to these Naa40p substrates being (at least partially) Nt-acetylated in the y[h*NAA40*] and y[h*NAA40^S^*] setups analyzed.

Although the from yeast to human conserved yH2A.Z and yH2A histones share 61% identity, yH2A.Z contains additional N- and C-terminal amino acids and comprises roughly 10% of the total cellular H2A which is almost exclusively incorporated at transcription start sites. Although not directly participating in the activation of transcription, H2A.Z seems to play a role in preventing DNA-methylation dependent chromatin silencing and chromatin accessibility [51].

Next to Lge1, we previously reported 3 other putative redundant yNaa40p/yNatA substrates Scl1 (SGAA), Ypi1 (SGNQ) and Dph1 (SGST). More specifically, their corresponding SG- starting N-termini also remained (partially) Nt-acetylated in yeast deleted for yNatA [17], an observation in line with Lge1, and H2A and H4 being Nt-acetylated in the same yeast background. Considering all 7 (highly likely) yNaa40p substrates, it is apparent that the consensus N-terminal sequence deviates from the P1′-P4′ SGSR consensus hNaa40p substrate specificity determined by proteome-derived peptide library screening [14] and structural analysis of *Schizosaccharomyces pombe* Naa40p (SpNaa40p) and hNaa40p bound to an H4/H2A N-terminal substrate [11]. More specifically, the yNaa40p substrate specificity appears confined to the P1′ and P2′ amino acids SG- viewing the divergence observed at P3′-P4′, thus indicating that the substrate specificity of yNaa40p is less extended when compared to hNaa40p and SpNaa40p substrate specificities, and that scNaa40p likely evolved to accommodate N-termini with varying P3′ and P4′ specificities. This observation is also in line with yH2A (SGGK) representing a yNaa40p-only substrate as yH2A is refractory towards hNaa40p mediated Nt-acetylation when ectopically expressed in yeast [14], a finding in contrast with yH4 (SGRG) representing a human and yeast Naa40p substrate [14]. Structural analysis of scNaa40p bound to an N-terminal substrate would potentially reveal structural differences in substrate recognition and differences in the extended substrate specificity profiles of orthologs Naa40p.

In addition to the Naa40p substrates of which note, H2A.X and SMARCD2 were also put forward as two putative hNaa40p substrates based on their SGRG- N-termini matching the Naa40p consensus substrate specificity [11]. In this context, it is interesting to note that our differential interactomics data obtained by BioID identified several known components of the multiprotein chromatin-remodeling SWI/SNF and related complexes—and thus known SMARCD2 interactors—(e.g., the catalytic subunits SMARCA4, SMARCA5, SMARCE1 and SIN3A) as significantly enriched in the proxeome of hNaa40^L^. Furthermore, and besides the known substrate histone H4, the core histone macro-H2A.1 (SSRG-) was also identified in the hNaa40^L^ interactome. Overall, our data indicates that known and putative hNaa40p substrates are specifically enriched in the interactome of nuclear localized hNaa40^L^, again pointing to possible post-translational nuclear Nt-acetylating activity of hNaa40^L^ over these (putative) substrates. However, it cannot be excluded that a small cytosolic resident fraction of hNaa40^L^ may also contribute to cytosolic hNaa40p activities.

BioID also further refined putative proteoform-specific roles of hNaa40p. As deduced from their respective interactomes, hNaa40^S^ likely represents the main co-translationally acting actor, while in line with its N-terminal extension holding a nucleolar localization signal, the hNaa40^L^ interactome was enriched for nucleolar proteins implicated in ribosome biogenesis and the assembly of ribonucleoprotein particles. Overall, our BioID data indicates a proteoform-specific segregation of previously reported Naa40p activities.

To create a selective h*NAA40^L^* knockout HAP1 clone, a guide RNA was designed to target exon 2 in the region preceding the dTIS. Since the effectiveness of gene deletions is frequently assessed at the gene level, it is striking that in roughly one third of the attempts to knock out genes via CRISPR-Cas9-induced frameshift mutations, translation products are still produced due to alternative splicing events skipping the CRISPR-induced frameshift or to translation initiation at a dTIS not impacted by the mutation [52]. The observed expression of residual hNaa40^S^ in the h*NAA40*^L^ line and our previous results on observed METAP2 activity in a so-called *METAP2* knockout line [53], are in line with the aforementioned study and indicates that besides the creation of full knockouts, CRISPR-Cas9 induced frameshifts can result in the expression of truncated proteoforms with (partially) preserved protein function. These observations warrant caution when determining gene essentiality based on CRISPR-Cas9 targetability and thus necessitating phenotypic interrogation of lines generated by CRISPR-Cas9. In the specific case of h*NAA40*, inspection of the h*NAA40^L-^* HAP1 line demonstrated that while the frameshift permitted hNaa40^S^ proteoform expression in the h*NAA40*^L^ background, residual hNaa40^S^ was deficient in preserving histone H4 Nt-acetylation capacity indicating that either the remaining hNaa40p^S^ expression levels were too low or alternatively, hNaa40^L^ is mandatory for (nuclear) histone H4 Nt-acetylation.

## 4. Materials and Methods

### 4.1. Cell Culture

The human HCT 116 colon cancer cell line was kindly provided to us by the Johns Hopkins Sidney Kimmel Comprehensive Cancer Center (Baltimore, BAL, MD, USA) and cultivated in McCoy’s medium (Gibco™, Thermo Fisher Scientific Inc., Waltham, MA, USA, cat n°22330-021). K-562 (ATCC, American Type Culture Collection, Manassas, VA; CCL-243) and HL-60 (ATCC, CCL-240) cells were grown in RPMI 1640 medium (Invitrogen, Life technologies, Waltham, MA, USA, cat n°61870-044). Human A-431 (epidermoid carcinoma; American Type Culture Collection (ATCC), Manassas, VA, USA; ATCC^®^ CRL-1555™), Human HeLa cells (epithelial cervix adenocarcinoma, American Type Culture Collection, Manassas, VA, USA; *ATCC*^®^ CCL-2™) and Flp-In™ T-REx™-293 cells (Thermo Fisher Scientific) were cultured in GlutaMAX containing Dulbecco’s Modified Eagle Medium (DMEM) (Invitrogen, cat n°31966047). The HAP1 wild type and CRISPR/Cas9 edited human knockout cell lines obtained from Horizon Genomics GmbH (Vienna, Austria) were grown in Iscove’s Modified Dulbecco’s Medium (IMDM) (Invitrogen, cat n°31980-048). More specifically, two h*NAA40* knockout clones (i.e., a h*NAA40*-4bp deletion knock out (HZGH C003224c001 (clone 1) and HZGHC003224c012 (clone 2)) containing a frameshift mutation in h*NAA40* were obtained (i.e., h*NAA40* knockout HAP1 clones carrying a 4bp deletion in the second coding exon of NM_024771 at genomic location chr11:63945863 were created using the guide RNA sequence GAAGAACAGAAGCGGTTGG leading to the corresponding h*NAA40* genomic sequence of GGAGAAGAAGCAGAAGCGGT(TGGA insertion)GGAGCGAGCAGCCATGGATG which was sequence verified by RNA-seq and Sanger sequencing (data not shown) upon PCR amplification from purified genomic DNA using the following primers; CTCTTTCTGAAGGGAAGCTGAGAAG (forward) and TTATAAACCAAGCTGAAGTCCCAGG (reverse) and sequencing primer CTCTTTCTGAAGGGAAGCTGAGAAG).

All media contained 2 mM alanyl-L-glutamine dipeptide (GlutaMAX™) and were supplemented with 10% fetal bovine serum (FBS) (HyClone, Gibco™, cat n°10270106, E.U.-approved, South American origin), 50 units/mL penicillin (Gibco™, cat n°15070-063) and 50 μg/mL streptomycin (Gibco™, cat n°15070-063). Parent Flp-In™ T-REx™-293 cells were additionally supplemented with 15 μg/mL blasticidin (InvivoGen, cat n°ant-bl-1) and 100 μg/mL zeocin for cultivation, while zeocin was added at 800 µg/mL for counter selection of stable transfected cells (InvivoGen, cat n°ant-zn1). Stable transformants were maintained in media supplemented with blasticidin and 50 μg/mL Hygromycin B Gold (InvivoGen, cat n°ant-hg-1).

For L-Arg SILAC (stable isotope-labeling by amino acids in cell culture) labeling of K-562 cells, cells were grown in RPMI 1640 SILAC medium (Invitrogen, cat n°61870-010) containing either natural, ^13^C_6_- or ^13^C_6_^15^N_4_-labeled L-Arg (Cambridge Isotope Laboratories, Andover, MA, USA) [24] at a concentration of 57.5 μM (5% of the suggested concentration in RPMI 1640 medium) and media were supplemented with 10% dialyzed fetal bovine serum (Invitrogen, cat n°26400-044). Cell populations were cultured for at least six population doublings for complete incorporation of the labeled L-Arg.

K-562 cells were induced to differentiate into megakaryocytes at a density of 3 × 10^5^ cells per ml using 40 nM PMA (Sigma-Aldrich, St. Louis, MO, USA) and collected at the indicated time points after induction. Light microscopic inspection of cell morphology to monitor cellular differentiation was performed on an Axiovert 25 microscope (Carl Zeiss, Jena, Germany).

All cells were cultured at 37 °C in a humidified incubator at 37 °C and 5% CO_2_ or 8% CO_2_ and passaged every 3–4 days.

### 4.2. Bacterial Strains

For cloning, *Escherichia coli* strains DH5α, DH10B or Top10 were used using standard chemical transformation protocols.

### 4.3. N-Terminal Proteomics

PMA treatment of L-Arg SILAC labeled K-562 cells proceeded for 4 or 6 days with 40 nM of PMA. Control cells were harvested on the day of PMA treatment initiation. Cells that became adherent during the differentiation process were detached using cell dissociation buffer (Invitrogen, cat n°13151014) and pooled with non-adherent cells. The obtained cell pellets (20 × 10^6^ cells) were washed twice in phosphate-buffered saline (PBS) and lysed in 1 mL of 50 mM HEPES pH 7.4, 100 mM NaCl, 0.8% CHAPS with a cOmplete™ protease inhibitor cocktail tablet added (Roche, cat n°11697498001) for 10 min on ice and centrifuged for 15 min at 16,000× *g* at 4 °C. Subsequently, the cleared lysates were subjected to N-terminal COFRADIC analysis as described previously [26] where in vitro ^3^C_2_D_3_-acetylation (using NHS (*N*-hydroxysuccimide) ^13^C_2_D_3_-acetate) was performed at the protein level to discriminate between in vivo free and in vivo Nt-acetylated protein N-termini, further enabling calculation of the extent of in vivo Nt-acetylation [54]. Differential L-Arg SILAC labeling moreover enables the relative quantification of Nt-peptide abundancy [25,26].

### 4.4. Generation of TIS Mutagenized hNAA40 and Flag-Tagged hNaa40p Expressing Constructs

A sequence-verified full-length I.M.A.G.E. cDNA clone of h*NAA40* (matching mRNA gi:13376118, cDNA clone MGC:59726, IMAGE:6292760)) cloned in the pOTB7 vector (IRAUp969D10104D, RZPD Imagenes, Germany) served as template for site-directed PCR-mutagenesis (QuickChange, Stratagene, La Jolla, CA, USA) according to the manufacturer’s instructions and as described in [55] and using the primer pairs (10 nmol, RP-cartridge Gold, Eurogentec) Naa40a120c forward: 5′-GGAGCGAGCAGCCCTGGATGCCGTTTG-3′ and Naa40a120c reverse: 5′-CAAACGGCATCCAGGGCTGCTCGCTCC-3′ to introduce an ATG to CTG mutation of the newly identified dTIS in the coding sequence of h*NAA40*, concomitantly resulting in recoding of Met22 to Leu.

pMET7 Flag-tagged hNaa40p expressing constructs encoding either wild type full-length and thus also potential short hNaa40p (hNaa40^L/S^), hNaa40p with mutation of the dTIS (hNaa40^L(dTIS>CTG)^) or mutation of the dbTIS (hNaa40^S(dbTIS>CTG)^), in addition to the shorter hNaa40^S^ coding sequence (CDS) were generated using standard ApaI and SalI-HF restriction cloning and ligation. The following PCR primer pairs were used to introduce the ApaI and SalI restriction sites (underlined sequence), wt. or mutagenized TIS (bold) and to amplify h*NAA40^L/S^* from the wild type h*NAA40* sequence (ApaI forward: 5′-GCGAGGGCCCAGCTT**ATG**GGGAGAAAGTCAAGC-3′), h*NAA40^L^*^/(dTIS>CTG)^ from dTIS > CTG mutagenized *NAA40* sequence (ApaI forward: 5′- -3′), h*NAA40*
^*S*(dbTIS>CTG)^ from the wild type h*NAA40* sequence (ApaI forward: 5′-GCGAGGGCCCAGCTT**CTC**GGGAGAAAGTCAAGC-3′), h*NAA40*^S^ from the wild type h*NAA40* sequence (ApaI forward: 5′- GCGAGGGCCCAGCTT**ATG**GATGCCGTTTGTGCC-3′) and all PCRs using the SalI primer 5′- GCGAGTCGACGTGGCAGCAGCCACC-3′ as reverse primer. The correctness of all (mutant) cDNA sequences was confirmed by Sanger sequencing.

### 4.5. Coupled In Vitro Transcription and Translation Assay

h*NAA40* and dTIS mutagenized h*NAA40* pOTB7 constructs were used as templates for in vitro coupled transcription/translation in a rabbit reticulocyte lysate system according to the manufacturer’s instructions (Promega, Madison, WI, USA) to generate [^35^S] methionine labeled translation products. 5 μL of the translate reaction was diluted 10-fold in 10 mM Tris pH 8.0. NuPAGE^®^ LDS Sample Buffer (Invitrogen, Cat n°NP0007) was added and the samples heated for 10 min at 70 °C. Samples were separated on 4–12% NuPAGE^®^ Bis-Tris gradient gels (1.0 mm × 12 well) (Invitrogen, Cat n°NP0321) using MOPS SDS Running Buffer (Invitrogen, Cat n°NP0001). Subsequently, proteins were transferred onto a polyvinylidene fluoride (PVDF) membrane, air-dried and exposed to a film suitable for radiographic detection (ECL (enhanced chemiluminescence) Hyperfilms, Amersham Biosciences, Buckinghamshire, UK) and radiolabeled proteins were visualized by radiography.

### 4.6. Yeast Cultivation and Strain Generation

Yeast was cultivated according to standard procedures. An N-terminal truncated h*NAA40* expression construct was made by subcloning the CDS part encoding h*NAA40^S^* from the I.M.A.G.E. cDNA clone (see above) to the pBEVY-U *S. cerevisiae* expression vector with bidirectional promoter using XmaI and EcoRI restriction cloning. The following PCR primer pair was used to introduce the XmaI and EcoRI restriction sites (underlined sequence) and to amplify h*NAA40^S^*; h*NAA40^S^* XmaI forward: 5′-GATCATCCCGGGCGCTATGGATGCCGTTTGTGCC-3′ and h*NAA40^S^* EcoRI reverse: 5′-GATCATGAATTCTCAGTGGCAGCAGCCACCAC-3′. The correctness of the construct was confirmed by Sanger sequencing. pBEVY-U-h*NAA40* [14] and pBEVY-U-h*NAA40^S^* expression vectors were introduced in the y*naa40*Δ *S. cerevisiae* (Y16202) *MAT* alpha strain (Euroscarf) (y*naa40*Δ), to generate a y[h*NAA40*] and y[h*NAA40^S^*] strain. To generate a control yeast strain pBEVY-U was introduced in the *S. cerevisiae* BY4742 (Y10000) WT *MAT* alpha strain [14]. Selection of transformed strains was performed on plates lacking uracil.

### 4.7. Histone Extraction Followed by In-Gel Stable-Isotope Labeling (ISIL) and In-Gel Digestion

For histone isolation, the equivalent of one 10 cm subconfluent HAP1 dish or alternatively an 80 mL yeast culture cultivated in Synthetic Defined (SD) Yeast medium lacking uracil (Sigma) to an OD_600_ of ~5.5–6.0 were used as input samples for performing histone extraction. Cell pellets collected by centrifugation (5 min at 1500× *g*) were resuspended in 3 mL (yeast) or 500 µL lysis buffer (0.2% TRITON-X-100, 10 mM HEPES (pH 7.6), 1.5 mM MgCl2, 10 mM KCl, 1 mM sodium orthovanadate, 10 mM sodiumbutyrate, 20mM β-glycerophosphate and a cOmplete™ ethylenediaminetetraacetic acid (EDTA) free protease inhibitor cocktail tablet added (Roche, cat n°11873580001). In case of HAP1 lysates, the lysates were left on ice for 10′ and centrifuged at 3000× *g* (4 °C) and the pellets kept for subsequent histone extraction (see below). For all yeast samples, an equal volume of 0.5 mm glass beads was added to the suspensions for 10 rounds of vortexing (30 s) and cooling on ice (30 s). After lysis, the glass beads were removed by centrifugation for 5 min at 3000× *g* and 4 °C and the cleared supernatant was centrifuged for 10 min. at 10,000×*g* and 4 °C and kept for Western blot analysis (see below). Pellets obtained in case of yeast and HAP1 samples were washed in lysis buffer followed by centrifugation for 5 min. at 3000× *g*. The pellets were incubated in 450 µL (yeast) or 150 µL (HAP1) 0.4 M HCl for 1 h and centrifuged for 15 min. at 13,000× *g*. 6 volumes of cold aceton (−20 °C) were added to the supernatant, vortexed and acid-extracted proteins precipitated overnight at −20 °C. After centrifugation for 15 min. at 13,000 × *g*, and 3 rounds of washing with cold aceton, the pellets were dried for 5 min in a vacuum concentrator. The dried pellets enriched for histones were dissolved in sample buffer (8 M urea, 5% β-mercaptoethanol and 10 mM Tris-HCl (pH 7.0)) and added sample loading buffer (Bio-Rad XT sample buffer, Bio-Rad Munich, Germany). The samples were separated on a 12% on a 12% gradient XT precast Criterion gel using XT-MOPS buffer (Bio-Rad) using XT-MOPS buffer (Bio-Rad) at 150–200 V. Coomassie-stained histone bands were cut from the gel, followed by in-gel-stable isotope labeling (ISIL) making use of *N*-hydroxysuccimide (NHS) ^13^C_2_D_3_-acetate and in-gel protein digestion (i.e., a 30 min. and 2 h. digestion at 37°C)as described previously [14,30]. The resulting peptide mixtures were acidified (0.1% formic acid) and analyzed by LC-MS/MS analysis as described previously [14].

### 4.8. Metabolic Labeling Using 12-Azidododecanoic Acid to Assess the N-Myristoylation Status of Transiently Expressed hNaa40p Proteoforms

Human HCT 116 cells seeded one day prior to transfection at 5.8 (in case of 24 h azidomyristate labeling) or 7.5 (8 h azidomyristate labeling) × 10^6^ cells/per 10 cm plate in 8 mL DMEM without FBS, were transiently transfected for 24 h with 9.36 µg of h*NAA40*-flag expression vectors and using 35 µL Fugene HD (Roche, cat. n°04709705001) according to the manufacturer’s instructions. More specifically, DNA/transfection reagent complexes in 500 µL Opti-MEM (Invitrogen, cat n°51985-042) were allowed to form by mixing and incubation at room temperature for 10 min. Medium was replaced with 5ml Opti-MEM and the DNA complexes (545 µL) were added to the medium by gently mixing. Cells were incubated at 37 °C and 5% CO_2_. After 4 h of transfection, 1100 µL 50% FCS containing DMEM was added to bring to a f.c. of 10% FCS and incubated at 37 °C and 5% CO_2_. 24 h post-transfection, the medium was replaced with 5 mL fresh medium and 50 or 100 μM 12-azidodo-decanoate (azidomyristate) (Click-IT™ Myristic Acid, Azide, ThermoFisher Scientific, Cat n° C10268) in dimethyl sulfoxide (DMSO) or vehicle was added to the media and incubated for an extra 8 or 24 h as indicated. The cells were washed with cold PBS, harvested in PBS by scraping, and lysed for 10′ on ice by resuspension of the cell pellets in 1 mL of lysis buffer (1% SDS in PBS) and 5′ of vortexing, followed by centrifugation of the lysates at 16,000× *g* (10 min at 4 °C) and collection of the cleared supernatant. Protein concentration of the supernatant was determined using the Bio-Rad DC Protein Assay according to the manufacturer’s instructions. Protein concentrations were adjusted to 600 µg/mL with lysis buffer and per sample and the CuAAC reaction performed essentially as described in [56].

In brief, a click mixture for CuAAC was prepared by the ordered addition and intermittent vortexing between each addition of; alkyn-biotin conjugate (PEG4 carboxamide-Propargyl Biotin, Invitrogen, Cat n° B10185), 14 μL of 5 mM stock solution DMSO, final concentration 0.0875 mM), CuSO_4_ (28 μL, stock solution 50 mM in DMSO, final concentration 1.75 mM) and tris(benzyltriazolylmethyl)amine (TBTA) (28 μL, stock solution 5 mM in DMSO, final concentration 0.175 mM). The click mixture (70 μL) was added to each sample to 700 µL of lysate (420 µg of protein), mixed, and sodium ascorbate additionally added (28 μL, stock solution 50 mM, final concentration 1.75 mM). The reaction mixture was left at room temperature (RT) for 1 h. Next, 7 mL ice-cold MeOH and EDTA (final concentration 10 mM) were added to each sample, vortexed and precipitated overnight at −80 °C. After overnight precipitation, the samples were centrifuged at 16,000× *g* (30 min at 4 °C) and the obtained protein pellets washed with 1 mL ice-cold MeOH and dried.

For enrichment of myristoylated proteins, pellets were resuspended in 2% SDS in PBS, 10 mM EDTA (140 μL). Once the pellet was completely dissolved, 560 μL of PBS was added (0.4% SDS f.c.). 100 µL was served as ‘input’ sample. The remaining was diluted to 0.2% SDS by the addition of 600 μL PBS. Dynabeads MyOne Streptavidin C1 (80 μL) (ThermoFisher Scientific, cat n°65601) washed 3 times with 1 mL 0.2% SDS in PBS were added to the sample and gently vortexed for 90 min. The supernatant was removed (unbound fraction) and the beads were washed 3 times with 0.2% SDS (3 × 1000 μL). 120µL of 2% SDS in PBS, 42 µL 4× Bio-Rad XT sample loading buffer and 8.5 µL 20× Bio-Rad XT reducing agent were added to the beads and boiled for 5 min. Samples were centrifuged at 1000× *g* for 2 min and loaded on an SDS–polyacrylamide gel electrophoresis gel (‘before pull-down’ sample: 20 μL; supernatant: 20 μL pull-down: 20 μL (5*more). Elution fractions after streptavidin purification were centrifuged at 1000× *g* for 2 min. 10 µg protein (input/unbound) and the equivalent of 50 µg protein input (elution) were analyzed on a 4–12% gradient XT precast Criterion gel using XT-MOPS buffer (Bio-Rad).

### 4.9. Total Cell Lysis, Cytoplasmic and Nuclear Fractionation

A-431 cells were transiently h*NAA40*-flag transfected for 24 h with h*NAA40*-flag expression vector and using Fugene HD (Roche) according to the manufacturer’s instructions and as described above. Cells were harvested by trypsin-EDTA or scraping in D-PBS without Ca^2+^ and Mg^+^ and collected by centrifugation for 5′ at 600× *g* at 4°C. The cell pellets were washed in ice-cold D-PBS, re-centrifuged and resuspended at ~5 × 10^6^ cells/mL in either lysis buffer (50 mM Tris-HCl pH 8.0, 150 mM NaCl, 1% NP-40) by repeated freezing and thawing for obtaining total lysates (TL) or alternatively, for cytoplasmic and nuclear fractionation in buffer A containing 10 mM HEPES, pH7.5, 10 mM KCl, 1 mM MgCl_2_, 5% glycerol, 0.5 mM EDTA, 0.1 mM EGTA supplemented with 0.5 mM DTT, 2 mM Pefablock and 0.15 U/mL aprotinin. For total lysates (TL), lysis was allowed to proceed for 15′ on ice and the supernatants collected after centrifugation for 10′ at 10,000× *g*. For cytoplasmic and nuclear fractionation, cell suspensions were left to swell for 15′ on ice in buffer A. Thereafter, NP-40 was added to a final concentration of 0.65% (*v*/*v*) and the cells lysed by vortexing for 10 s. Remaining intact nuclei were pelleted by centrifugation for 10′ at 700× *g*, the supernatant collected (cytoplasmic fraction (C)) collected and the nuclear-enriched fractions re-centrifuged at higher speed (5′ at 5000× *g*) for complete removal of residual buffer A, and followed by two washes in buffer A containing 1% NP-40. The obtained pellets enriched for nuclei were lysed in lysis buffer (50 mM Tris-HCl pH 8.0, 150 mM NaCl, 1% NP-40) by repeated freezing and thawing. Supernatants enriched for nuclear proteins (N) were collected by transfer of the supernatant obtained after centrifugation at 16,000× *g* for 20 min at 4 °C. Protein concentration was performed by Bio-Rad DC Protein Assay determination according to the manufacturer’s instructions and equal protein amounts of cytoplasmic and nuclear fractions were analyzed for the comparison of a protein’s cytoplasmic versus nuclear distribution within the cell.

### 4.10. SDS-PAGE and Immunoblotting

Sample loading buffer (Bio-Rad XT sample buffer) and reducing agent (Bio-Rad) was added to the samples according to the manufacturer’s instruction and equivalent amounts of protein (30 µg as measured using the DC Protein Assay Kit (Bio-Rad) and proteins were separated on a 4 to 12% on a 12% gradient XT precast Criterion gel using XT-MOPS buffer (Bio-Rad) at 150–200 V. Subsequently, proteins were transferred onto a PVDF membrane. Membranes were blocked for 30 min in a 1:1 Tris-buffered saline (TBS)/Odyssey blocking solution (cat n° 927-40003, LI-COR, Lincoln, NE, USA) and probed by Western blotting. Following overnight incubation of primary antibody in TBS-T/Odyssey blocking buffer and three 10 min washes in TBS-T (0.1% Tween-20), membranes were incubated with secondary antibody for 30 min in TBS-T/Odyssey blocking buffer followed by 3 washes in TBS-T or TBS (last washing step). The following antibodies were used: mouse anti-Flag (Sigma, F3165; 1/5000), mouse anti-GAPDH (Abcam, ab9484, 1/10,000), rabbit anti-histone H3 (Cell Signaling, #9715, 1/2000), rabbit anti-hNaa40p/anti-nat11 (Sigma, SAB3500167, 1/1000) (polyclonal antibody raised against 15 aa at the C-terminus of hNaa40p), rabbit anti-α-Tubulin (Cell Signaling, #2144, 1/2000), mouse anti-α-Tubulin (Sigma, T5168, 1/1000), rabbit anti-PRKACA (Cell Signaling, #5842S, 1/2000), rabbit anti-c-Src (Cell Signaling, #2123, 1/1000), streptavidin-Alexa Fluor™ 680 Conjugate (Invitrogen, S32358, 1/10,000), anti-mouse (IRDye 800 CW goat anti-mouse antibody IgG, LI-COR, cat n° 926-32210, 1/10,000) and anti-rabbit (IRDye 800 CW goat anti-rabbit IgG, LI-COR, cat n° 926-3221, 1/10,000). The bands were visualized using an Odyssey infrared imaging system (LI-COR).

### 4.11. Generation of Stable, Inducible Expression Cell Lines of C-Terminal BirA*-Tagged HNaa40p Proteoforms

*NAA40^L^*^(dTIS>CTC)^ and *NAA40^S^* expressing constructs for BioID were generated via Gateway cloning. More specifically, Flag-tagged h*NAA40*
^*L*(dTIS>CTC)^ and h*NAA40* vectors served as templates to generate attB-flanked h*NAA40*
^*L*(dTIS>CTC)^ and h*NAA40^S^* PCR products suitable for use in a Gateway^®^ BP recombination reaction with a donor vector (pDONR221, Invitrogen, cat n°12536-017) thereby creating an entry clone. Forward primers of h*NAA40**^L^*^(dTIS>CTC)^ (5′- GGGGACAAGTTTGTACAAAAAAGCAGGCTTCACCATGGGGAGAAAGTCAAGCAAAGCC-3′) and h*NAA40^S^* (5′-GGGGACAAGTTTGTACAAAAAAGCAGGCTTCACCATGGATGCCGTTTGTGCCAAAGTGGACG-3′) and the respective reverse primer (5′-GGGGACCACTTTGTACAAGAAAGCTGGGTCGTGGCAGCAGCCACCACAGTG-3′) was used to generate attB-flanked PCR products. The reverse primer was designed to fuse the desired PCR products in frame with a C-terminal BirA*-FLAG tag encoded by the destination vector. To create h*NAA40*
^*L*(dTIS>CTC)^ and h*NAA40^S^* expression clones, the inserts of the entry vectors were recombined into the into the pDEST 5′ BirA*-FLAG-pcDNA5-FRT-TO destination vector (Invitrogen, cat n°12285-011) using LR-clonase (Invitrogen, cat n° 11791-020) according to the manufacturer’s instructions. All constructs were sequence verified by Sanger sequencing. Stable cell lines expressing hNaa40^L^ or hNaa40^S^ proteoform bait proteins in Flp-In T-REx-293 cells that contain a stably integrated flippase recognition target (FRT) site at a transcriptionally active genome locus (Thermo Fischer Scientific) were generated as cell pools as described in [57]. Briefly, Flp-In T-REx 293 cells were co-transfected with a h*NAA40*
^*L*(dTIS>CTC)^ or h*NAA40^S^* expression construct and the Flp recombinase encoding pOG44 plasmid (Invitrogen) in a 1:9 ratio (in 6-well plates at 60–70% confluency) using Lipofectamine^®^LTX with Plus™ Reagent (Invitrogen) in Opti-MEM according to the manufacturer’s instructions. Successful Flp-FRT-mediated recombination of the h*NAA40* inserts were selected for with 15 μg/mL blasticidin and 50 μg/mL hygromycin B and the selection medium replaced every 2–3 days until the appearance of visible foci. Selected cell populations were pooled thereafter by incubation with cell dissociation buffer, collection by centrifugation (1000× *g*, 5 min) at 4 °C and resuspension and maintenance in complete DMEM supplemented with blasticidin and 50 μg/mL hygromycin B. (moreover Pen./Strep.) Cell pools were expanded and grown without tetracycline. Stable cells expressing BirA*-FLAG fused to enhanced green fluorescent protein (eGFP) were used as negative control setup for the BioID experiments and processed in parallel to the hNaa40p proteoform bait proteins.

### 4.12. BioID

For BioID, stable cell lines were plated at 1.5 × 10^7^ cells (corresponding to 70–80% confluence) per 100-mm plate. 6 h after plating, expression was induced by the addition of an optimized concentration of the stable tetracycline derivative doxycycline (DOX) at 2 ng and 5 ng/mL for hNaa40p proteoform and eGFP expression, respectively, and 16 h thereafter, 50 μM biotin was added for an extra 24 h. For BioID, per construct, six 100-mm plates were used for harvesting. Cells were washed with D-PBS and harvested by scraping and collection of the cells in 5 mL ice-cold PBS, followed by pelleting of the cells (in 2 aliquots containing respectively 10% and 90% of the cell material) at 600× *g* for 5′ at 4 °C. Pellets were kept frozen at −80 °C until further processing.

The frozen pellets corresponding to a total of ~1 × 10^8^ cells (90% of cell material) were resuspended in ~7.5 mL BioID lysis buffer (100 mM Tris pH 7.5, 150 mM NaCl, 2% SDS and 8M urea) to reach a protein concentration of ~2 mg/mL (i.e., the volume of BioID lysis buffer added was based on protein concentration determination using the DC Protein Assay Kit from Bio-Rad on the 10% aliquot samples lysed in 50 mM Tris-HCl pH 8.0, 150 mM NaCl and 1% NP-40). The suspensions were subjected to mechanical disruption by 3 repetitive freeze-thaw cycles in liquid nitrogen followed by 3 sonication cycles (i.e., 2 min of sonication on ice for 25-s bursts at output level 4 with a 40% duty cycle (Branson Sonifier 250; Ultrasonic Convertor)). To 7.5 mL of lysate (~15 mg of protein), a 500 µL 50% suspension of 3 times pre-washed streptavidin agarose beads (Novagen, Cat. N°69203) (i.e., washes were performed by repeated resuspension and pelleting of the beads by centrifugation at 600× *g* for 5 min with 2 mL of high stringency buffer; 100 mM Tris pH 7.5, 2% SDS, 8M urea and 150 mM NaCl) was added and incubated overnight at room temperature in a rotator to capture biotinylated proteins. After overnight incubation, the beads were pelleted (600× *g*, 2 min) and the unbound supernatant removed and kept for further analysis. The beads were then extensively washed (4 times 5′ and an additional 30′ wash with 1 mL of high stringency buffer (5 washes in total), followed by an additional 30′ wash with high salt buffer (1M NaCl, 100 mM Tris.HCl pH 7.5), a wash with ultrapure water (5′), and 3 × 1 mL 50 mM ammonium bicarbonate (pH 8.0) washes (5′). Following the final wash, the beads were pelleted, and any excess liquid was aspirated off. 90% of the beads were then resuspended in 600 µL of 50 mM ammonium bicarbonate (pH 8.0), and 1 μg of mass spectrometry grade trypsin (Promega, Madison, WI) was added. The samples were incubated overnight at 37 °C with vigorous mixing (850 rpm) to keep the beads in suspension. After overnight digestion, an additional 0.5 μg of trypsin was added, followed by an additional incubation of 2 h. The beads were pelleted (600× *g*, 2 min) and the supernatant was transferred to a fresh low protein binding tube (Eppendorf). The beads were washed with 2 × 300 μL HPLC (high performance liquid chromatography) grade water and the washes combined with the original supernatant. The peptide solution was acidified with 10% formic acid to reach a final concentration of 0.2% and cleared from insoluble particulates by centrifugation for 15 min at 16,000× *g* (4 °C) and the supernatant transferred to clean tubes. The samples were vacuum-dried in a SpeedVac concentrator, re-dissolved in 25 µL of 2 mM tris(2-carboxyethyl)phosphine in 2% acetonitrile, centrifuged for 15 min at 16,000× *g* (4 °C) and transferred to clean mass spec vial for LC-MS/MS analyses.

### 4.13. LC-MS/MS Analysis and Data Analysis of BioID Samples

BioID samples were analyzed by LC-MS/MS using an UltiMate 3000 RSLC nano HPLC (Dionex) in-line connected to a Q Exactive instrument (Thermo Scientific, Bremen, Germany) as previously described [58,59]. Raw data files were searched with MaxQuant [60] using the Andromeda search engine [61] (version 1.5.3.30) and MS/MS spectra searched against the Swiss-Prot database (taxonomy *Homo sapiens*) complemented with the BirA*Flag and eGFP sequences. Potential contaminants present in the contaminants.fasta file that comes with MaxQuant were automatically added. A precursor mass tolerance was set to 20 ppm for the first search (used for nonlinear mass recalibration) and set to 4.5 ppm for the main search. As enzyme specificity, trypsin was selected and up to two missed cleavages were allowed. Methionine oxidation and N-terminal protein acetylation were set as variable modifications. The false discovery rate for peptide and protein identification was set to 1%, and the minimum peptide length was set to 7. The minimum score threshold for both modified and unmodified peptides was set to 40. The match between runs function was enabled and proteins were quantified by the MaxLFQ algorithm integrated in the MaxQuant software [42]. Here, a minimum of two ratio counts and only unique peptides were considered for protein quantification.

For basic data handling, normalization, statistics and annotation enrichment analysis we used the freely available open-source bioinformatics platform Perseus (version 1.6.5.0) [62]. For data visualization we additionally made use of GraphPad Prism version 9.0.0. Perseus was used for non-supervised hierarchical clustering and 1D annotation enrichment with a two-sided test with Benjamini–Hochberg FDR correction (FDR = 0.01). Data analysis after uploading the protein groups file obtained from MaxQuant database searching was performed as described previously [59]. In brief, all replicate samples were grouped and LFQ-intensities log(2) transformed. Proteins with less than three valid values in at least one group were removed and missing values were imputed from a normal distribution around the detection limit (with 0.3 spread and 1.8 down-shift). Then, a multiple sample ANOVA t test (FDR = 0.01, S0 = 0.1) was performed to detect enrichments in the different BioID setups.

## Figures and Tables

**Figure 1 ijms-22-03690-f001:**
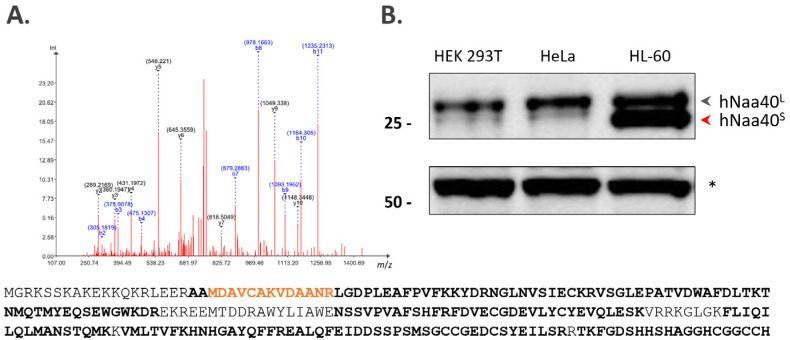
Identification and expression analysis of a novel N-terminally truncated hNaa40p proteoform, hNaa40^S^. (**A**) Mass spectrometric (MS) identification of an Nt-truncated Naa40p proteoform (hNaa40^S^) raised upon translation initiation at a downstream translation initiation start site (dTIS) as demonstrated by the MS^2^-based identification of the Nt-acetylated peptide MDAVCAKVDAANR (orange) matching amino acid sequence positions 22 to 34 of full-length canonical hNaa40^L^ (Q86UY6, length: 237 AA). The MS^2^ spectrum matches a doubly charged precursor mass of *m*/*z* 762.3546, corresponding to the modified peptide sequence Ac-M < Mox > DAVC < Cmm > AK < AcD3 > VDAANR, and with ‘Ac-’, ‘Cmm’ and ‘AcD3’ denoting an acetyl, carbamidomethyl and (trideutero)acetyl moiety [21]. Tryptic peptide identifications reported in ProteomicsDB ([22], accessed January 2021) are indicated in bold in the hNaa40^L^ protein sequence overall providing a sequence coverage of 80.17% and with the reported AAMDAVCAK peptide indicative of hNaa40^L^ expression (Appendix A) (**B**) Expression profiling of endogenous hNaa40p in different human cell lines by Western blot analysis indicates expression of hNaa40^L^ in all 3 cell lines and prominent expression of the short proteoform of hNaa40p (hNaa40^S^) in the pro-myelocytic leukemia cell lines HL-60. A non-specific band (*) serves as loading control.

**Figure 2 ijms-22-03690-f002:**
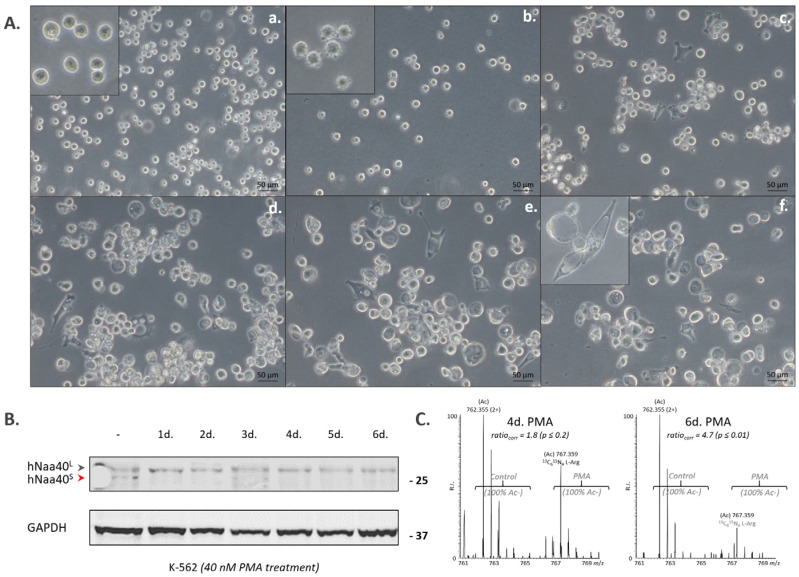
Differentiation of K-562 cells into megakaryocytes induces specific down-regulation of hNaa40^S^. (**A**) Morphology of phorbol 12-myristate 13-acetate (PMA) differentiated K-562 cells. Control K-562 cells (**a**), K-562 cells treated with PMA (40 nM) for 5 h, 1 day (d.), 2 d., 3 d. and 4 d. respectively (**b**–**f**) shows hallmarks of megakaryocytic differentiation upon PMA treatment. Images are annotated with a 50 µm scale bar. (**B**,**C**) Expression profiling of endogenous hNaa40p proteoforms at various time points of PMA treatment indicates specific down-regulation of hNaa40^S^ in PMA-differentiated K-562 cells by Western blot analysis (with GAPDH ((Glyceraldehyde-3-phosphate dehydrogenase), 36 kDa) serving as loading control) (**B**), and following the identification by N-terminal proteomics (**C**). Here, composite MS spectra of the Nt-peptide originating from hNaa40^S^ (MDAVCAKVDAANR) (see also Figure 1) in the control (light L-Arg SILAC (stable isotope-labeling by amino acids in cell culture) label) and 4 and 6 days PMA treated setups (heavy ^13^C_6_^15^N_4_ L-Arg SILAC label) are shown. In both setups, in vitro ^3^C_2_D_3_-acetylation was performed at the protein level in combination with differential L-Arg SILAC labeling [24] which allows for the calculation of the extent of Nt-acetylation as well as the relative quantification of Nt-peptide abundancy [25,26]. Overall, the N-terminus of hNaa40^S^ was fully Nt-acetylated in all setups analyzed and the relative concentration of hNaa40^S^ lowered after 4 d. (*p* ≤ 0.2) and 6 d. following PMA treatment (*p* ≤ 0.01).

**Figure 3 ijms-22-03690-f003:**
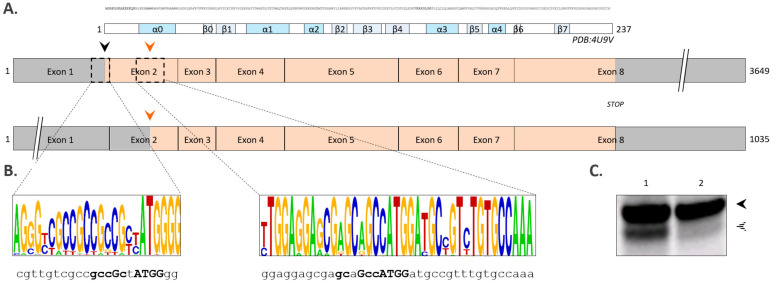
Alternative splicing or alternative translation can give rise to hNaa40^S^ expression. (**A**) Scheme depicting two alternatively spliced h*NAA40* transcripts; being transcript ENST00000377793.9 (3649 bp, transcript expected to code for the main functional isoform with transcript support level 1 according to the APPRIS database (http://appris.bioinfo.cnio.es, accessed on 1 December 2020)) and transcript ENST00000542163.1 (1035 bp, transcript support level 2). Exons are indicated as boxes and corresponding non-coding and coding regions colored grey and orange, respectively. When considering the longest transcript, hNaa40^L^ and hNaa40^S^ translation initiates at the first (black arrow) and second in-frame AUG codon (orange arrow), respectively. Exon 1 encodes for the first 2 amino acids of hNaa40^L^. The protein sequence of hNaa40^L^ (aa 1-237) with an indication of corresponding secondary structure elements of structure PDB:4U9V is provided above the transcripts. (**B**) Codon alignment of representative mammalian genomic sequences (alignment set hg38_58) in the vicinity of the canonical, database-annotated AUG TIS (dbTIS) (matching the human chr11:63939079-63939102 sequence indicated below) and the downstream AUG TIS (dTIS) (chr11:63945879-63945917 in human) of the *NAA40* locus using WebLogo. The heights of the nucleotides are indicative for their degree of conservation at the indicated position [27]. (**C**) Autoradiograph showing translation initiation of in vitro transcribed h*NAA40* at the AUG start codons encoding M1 (black arrowhead) and M22 (dashed arrowhead). For this, wild type (1) and a dTIS mutagenized (mutation of ATG encoding M22 to CTG (dTIS > CTG)) h*NAA40* pOTB7 constructs were in vitro transcribed and translated, and radiolabeled proteins visualized by radiography following sodium dodecyl sulfate (SDS)-PAGE and electroblotting.

**Figure 4 ijms-22-03690-f004:**
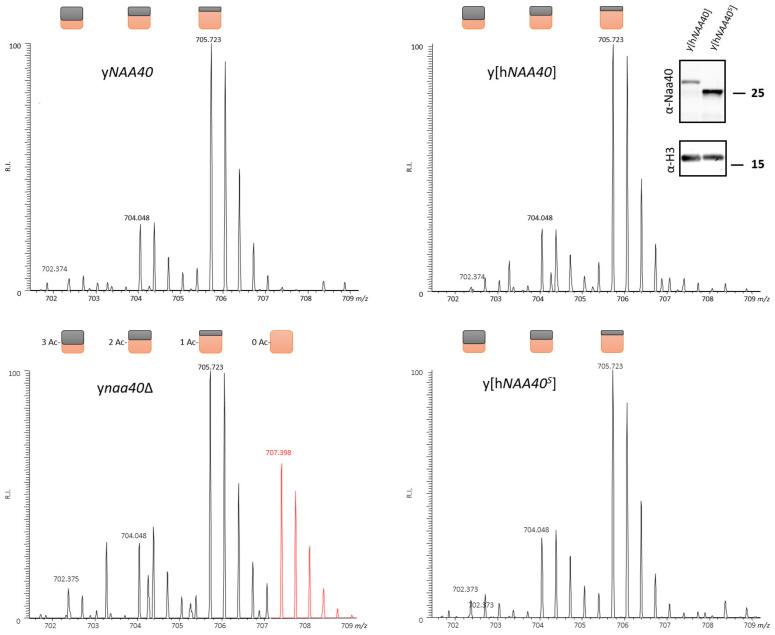
Ectopic expression of hNaa40^L^ and hNaa40^S^ in y*naa40*Δ yeast rescues *N*-acetylation of the newly identified in vivo yNaa40p substrate H2A.Z. Besides the identification of various *N*-acetylated forms of the mature N-terminus of yH2A.Z (^2^SGKAHGGKGKSGAKDSGSLR^21^), the fully *N*-free form (red MS trace, ‘0 Ac’ peak indicative of ‘no in vivo *N*-acetylation) was only demonstrated in the y*naa40*Δ setup as inferred from the triply charged variant observed at the higher *m*/z 707.398. The 5 Da spacing between the distinct isotopic envelopes, corresponds with the gain/loss of an in vivo *N*-acetyl group (i.e., free amines were in vitro *N*-acetylated using ^13^C_2_D_3_-NHS acetate; introducing a 5 Da heavier acetyl moiety to each free amine) [29,30]. Colored squares indicate the number of in vivo (black) and in vitro (orange) acetyl moieties present per detected isotopic envelope. Although y*naa40* deletion reduces the overall degree of H2A.Z *N*-acetylation (bottom left panel), ectopic expression of hNaa40^L^ as well as hNaa40^S^ restores the degree of *N*-acetylation of the H2A.Z N-terminus to wild type levels. The inset in the upper right panel confirmed the specific expression of hNaa40p proteoforms hNaa40^L^ and hNaa40^S^ in the y[h*NAA40*] and y[h*NAA40*^S^] strain, respectively. Histone H3 was probed as a loading control.

**Figure 5 ijms-22-03690-f005:**
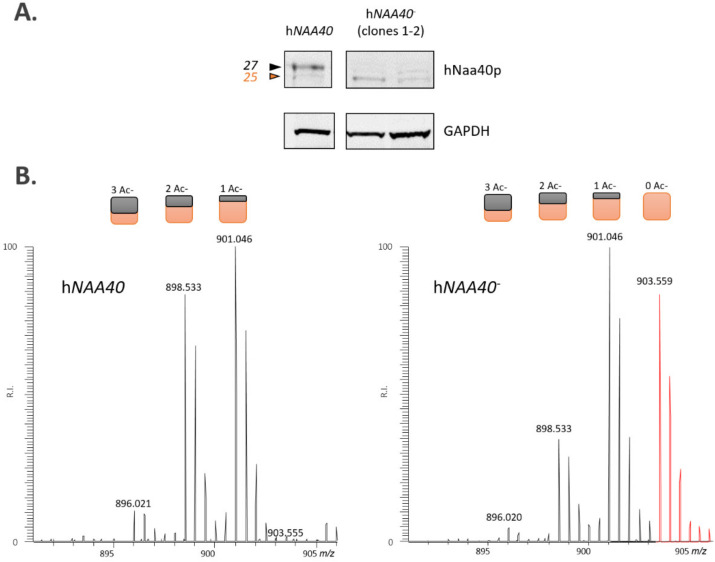
Residual hNaa40^S^ proteoform expression in h*NAA40^L-^* knockout HAP1 cells is deficient in maintaining Nt-acetylation of the hNaa40p substrate histone H4. (**A**) Western blotting analyses of corresponding HAP1 WT and h*NAA40^L-^* knockout HAP1 cell lysates (originating from two independent h*NAA40* knockout clones) immunoblotted with antibodies specific for hNaa40p and GAPDH are shown. (**B**) Besides the identification of various *N*-acetylated forms of the mature N-terminus of human histone H4 (^2^SGRGKGGKGLGKGGAKR^18^), the fully *N*-free form (red MS trace) was only demonstrated in the h*NAA40*^-^ setup as inferred from the doubly charged variant observed at a higher *m*/z (903.559). The 5 Da spacing between the distinct isotopic envelopes, corresponds with the gain/loss of an in vivo N-acetyl group (i.e., free amines were in vitro *N*-acetylated using ^13^C_2_D_3_-NHS acetate; introducing a 5 Da heavier acetyl moiety to each free amine [29,30]). Colored rectangles indicate the number of in vivo (black) and in vitro (orange) acetyl groups present per detected isotopic envelope.

**Figure 6 ijms-22-03690-f006:**
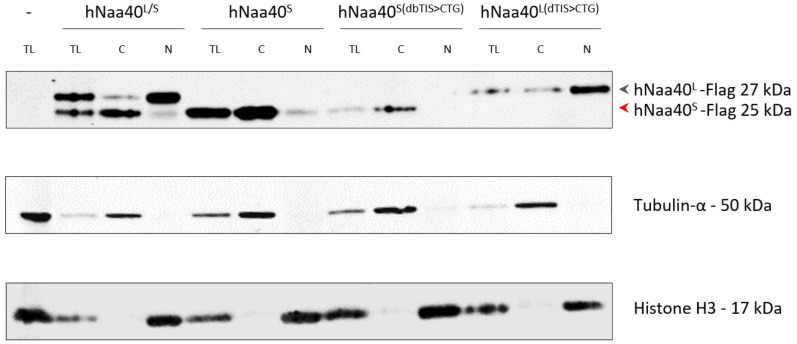
Differential subcellular localization of hNaa40p proteoforms. A-431 cells were transiently transfected with various C-terminally Flag-tagged hNaa40p expressing constructs encoding either both hNaa40p (hNaa40^L/S^), or solely the long (hNaa40 ^L(dTIS>CTG)^) or short (hNaa40 ^S(dbTIS>CTG)^ and hNaa40^S^) hNaa40p proteoform(s). Subcellular fractions enriched for nuclear (N) and cytoplasmic (C) proteins as well as total protein lysates (TL) were immunoblotted with antibodies specific for Flag, Tubulin-α and Histone H3, the latter two proteins serving as cytoplasmic and nuclear markers, respectively.

**Figure 7 ijms-22-03690-f007:**
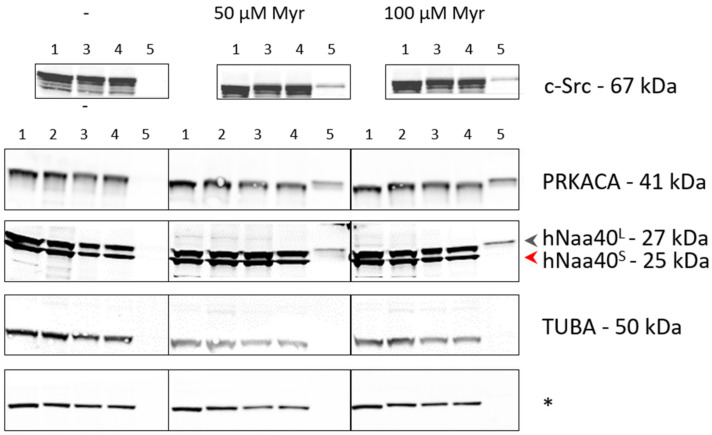
Azidomyristoylation of transiently expressed hNaa40^L^ in HCT 116 cells. (Transiently hNaa40^L/S^ expressing) HCT 116 cells were metabolically labeled for 8 h with 50 or 100 µM azidomyristate or left untreated (-). Besides the well characterized NMT substrates PRKACA and c-Src, a specific enrichment of hNaa40^L^ could be observed in the eluates (lanes 5) obtained by streptavidin-aided affinity purification in case of the azidomyristate treated samples (middle and right panels). Tubulin-α (TUBA) served as negative control and * indicates a non-specific loading control. 1 to 5 correspond to equivalents of the input lysate, clicked lysate, clicked + precipitated lysate, supernatant after streptavidin pull-down and eluate, respectively.

**Figure 8 ijms-22-03690-f008:**
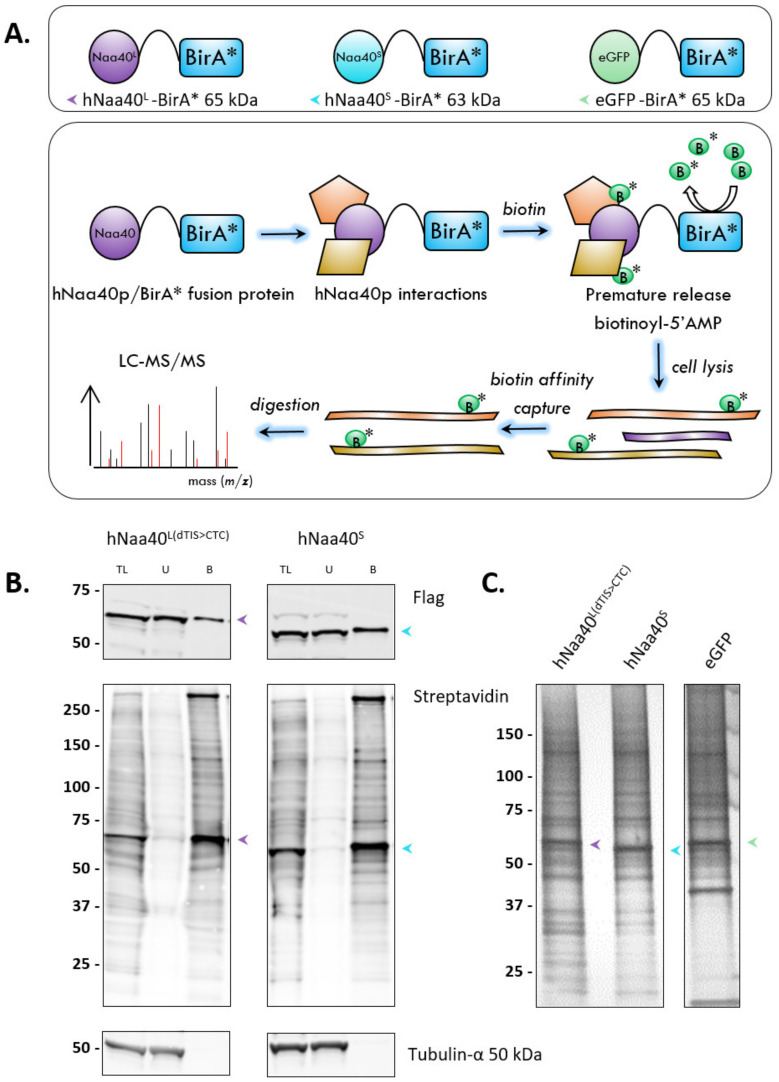
Proximity-dependent biotin identification (BiolD) for the discovery of differential hNaa40p proteoform interactors in Flp-In T-REx 293 cells. (**A**) Overview of the BiolD (setups) and MS analysis workflow. Biological replicate samples were prepared for all 3 setups analyzed (i.e., biological replicate samples of stable cell lines with doxycycline (DOX) induced expression of eGFP, hNaa40^L^ and hNaa40^S^ BirA* fusion proteins) (**B**) Western blot analysis showing expression and (partial) biotinylation of hNaa40^L^ and hNaa40^S^ BirA* fusion proteins, and specific enrichment of biotinylated hNaa40p proteoforms following streptavidin affinity capture. TL = total lysate, U = unbound fraction and B = bound fraction. Purple, light blue and green arrows indicate hNaa40^L^, hNaa40^S^ and eGFP-BirA* fusion proteins, respectively. The equivalents protein input of 50 and 200 µg ‘TL’ was analyzed for ‘TL’ and ’U’, and ‘B’ fractions, respectively. (C) Silver staining of biotinylated proteins following streptavidin purification (representative gel is shown). ~1.5% of total eluate sample corresponding to the input material of ~10^6^ cells was analyzed.

**Figure 9 ijms-22-03690-f009:**
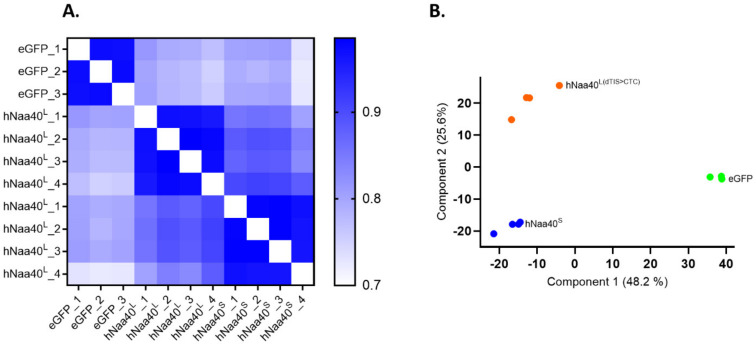
Heat map visualization of pairwise LFQ correlations (Pearson) (**A**) and PCA (**B**) plot of the variability between the different setups and replicate BioID samples analyzed. Green, blue and orange indicate eGFP, hNaa40^L(dTIS>CTC)^ (in A. shortly referred to as hNaa40^L^) and hNaa40^S^ setups, respectively.

**Figure 10 ijms-22-03690-f010:**
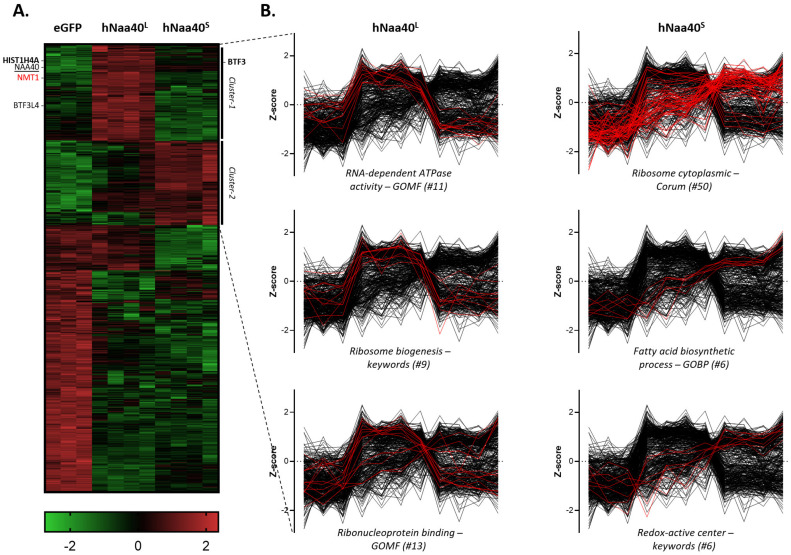
Proximity-dependent biotin identification (BiolD) enables the discovery of differential hNaa40p proteoform interactors. (**A**) Heat map representation of cluster analysis after ANOVA. The intensities of proteins with significantly different abundance (*p* ≤ 0.01) in the respective interactomes (#937) are represented. Two clusters (cluster 1 and 2) enriched for hNaa40^L^ and Naa40^S^ interactors can be observed. Green indicates low intensities while red indicates high intensities. (**B**) Profile plots of representative protein categories with significant increased abundance (false discovery rate (FDR) < 2%) in the hNaa40^L(dTIS>CTC)^ (middle panel) and hNaa40^S^ (right panel) setups are shown for the two indicated clusters (see also Appendix A) across all replicate samples and setups analyzed.

## Data Availability

Data is contained within (Supplementary Materials of) the article.

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
