# Peer review of "N-Terminal Acetyltransferase Naa40p Whereabouts Put into N-Terminal Proteoform Perspective"

_ijms, 2021, doi:10.3390/ijms22073690_

Round 1

Reviewer 1 Report

Naa40p is among the most selective N-terminal acetyltransferase (NAT) complexes identified .The present study deals with the identification of a Nt-truncated Naa40p proteoform (Naa40S).The newly identified shorter proteoform is functionally characterized in terms of enzymatic activity and subcellular localization pattern.

Specific points

- Figure 4-The authors demonstrate that both hNaa40L and hNaa40S are enzymatically active. Does it mean that both proteoforms recognize the same substrates in the cytosol and the nuclear enriched fraction. Why such a redundant activity.

- Cytoplasmic and nuclear-associated compartment fractionation reveal a clear hNaa40L enrichment in the nuclear enriched fraction while hNaa40S displays a significant cytoplasmic localization. However this evaluation is still qualitative and one cannot rule out the possibility that a (even small) fraction of  hNaa40L is present in the cytosol and  would contribute to  the enzymatic activity associated to hNaa40S.

- Nuclear localized hNaa40L is N-terminally myristoylated. Does myristoylation control the enzymatic activity.

- The authors do not provide any experimental arguments proving that C-terminal sequence might impact hNaa40p localization.

- Subcellular experiments are carried out with A-431 cell lines;azidomyristotylation is analysed on HCT-116 cell lines. Such a diversity of cells does not allow to draw final and convincing conclusions. This choice should be at least justified convincingly

Reviewer 2 Report

In the manuscript Big fish, little fish: N-terminal acetyltransferase Naa40p proteoforms caught in the act, the authors set out to study two proteoforms of Naa40p, a long and short form, to determine differences in activity and/or localization between the two.  While the experimentation on localization is convincing enough, a lack of clarity in explaining the activity assays (specifically Figures 4 & 5) severely hurt the approachability of this paper, and leave this reviewer unconvinced about the conclusions drawn from this paper.  That being said, there is a significant amount of experimentation present, and if the explanation of these figures were improved, or an improvement is made to the figures and/or legends themselves, it seems likely that this paper could be made ready for publication.  Specific issues are outlined below.

Moderate concerns:

Figure 4: This figure / figure legend is very difficult to follow.  So, the title states “Ectopic expression of hNaa40L and hNaa40S in ynaa40-∆ yeast rescues Nt-acetylation of the newly identified in vivo yNaa40p substrate H2A.Z.”  Is this based on the fact that in the ynaa40-∆ strain, we see the appearance of the 0 Ac peaks, which themselves are indicative of no in vivo acetylation?  And it disappears with ectopic expression of the short or long form of hNAA40? 

If this interpretation is correct, it could certainly stand to be stated more clearly.  If this interpretation is wrong, then a better explanation of how you reached your conclusion is needed.  Since Figure 5 also uses a similar experiment / logic, it is all the more important that the interpretation of Figure 4 is made clearer.

You state in lines 287-289, “Intriguingly, in the hNAA40L- HAP1 cell line expressing residual hNaa40S, it was observed that the Ser-Gly-Arg-Gly- H4 N-terminus remained Nt-free (Fig. 5).”  Are you stating there is no N-terminal acetylation?  Because while your figure clearly shows that while a 0 Acetylation population has appeared, there’s still a 1, 2 and 3 acetylation population as well.  How can you be sure none of these contain an N-terminal acetylation?

You state: “Further, we assessed the myristoylation status of transiently expressed hNaa40p and found that while hNaa40S remained unlabeled, hNaa40L was (partially) N-myristoylated and thus identified as 356 novel NMT substrate (Fig. 7 and data not shown).”  This conclusion, specifically the distinction between S remaining unlabeled while L was partially N-myristoylated, does not seem apparent from Figure 7.  Once again, this may have to do with an issue of clarity with how your figures are labeled / presented.

Overall, it’s worth pointing out that this paper is simply dense with jargon.  Given the nature of your experiments, it’s difficult to judge whether anything could be done to address this point, but the wall of abbreviations and esoteric language will present a barrier for the reader in its present form.

Finally, I will point out that, while I appreciate the desire to create a creative title for papers, I don’t feel, in this case, stating that you’ve “caught something in the act” is appropriate.  That would imply some sort of live imaging, which is not the case for this manuscript.  I would suggest a title that doesn’t overstate the findings.

Minor concerns:

Lines 76-85 of the introduction don’t appear to have a citation.

Partial list of typographical errors:

Line 247: Pursue should be Pursuit

Line 456: This needs to be cleaned up: “Data from the to date only handful of proteoforms functionally characterized however suggest…”

Reviewer 3 Report

The authors provide a very interesting article related to the dual proteoforms of the Naa40p in human. This is not completely new since similar results (for mice) have been previously observed and detailed in a dedicated article. BTW, this article is providing some new interesting facts such as N-Myristoylation of the hNaa40p(L) and nucleus vs cytosol location. N-terminal Myr is well known to be associated to specific location (especially at the membranes). Despite this manuscript is clear and associated with convincing results, I would suggest and require some modifications and updates to clarify a few points.

One of them is associated to Figure1. The hNaa40p full length sequence is provided from N to C-terminus with the characterized peptides in bold. Although it is interesting to see that the results are well in agreement with the previously characterized peptides displayed in the PeptideAtlas, I remain confused with the characterized peptide stating at position 20 since the proposed alternative start is at position 22. I strongly recommend providing the MS/MS spectra of both peptides since one of them confirm the long hNaa40p(L) sequence whereas the second confirm the alternative start hNaa40p(S).

Page 5, 2nd paragraph, the authors suggest that the alternative start, although there is a clear spliced transcript that validate the shorter form, could also be the result of an alternative TIS. They justify this option with the Kozac consensus pattern of this protein version. However, alternative starts are usually restricted to the first few residues (usually less than the 10 first residues). For hNaa40p(S), the start position will be at 20 residues from the expected primary start position that strongly decrease the probability of such event. If the authors would like to maintain this option, they should further provide some strong validation to this hypothesis.

Page 7 and figure 4: The authors provide some interesting results. BTW, due to the four lysines present in the peptide sequence which appears to be also K-Ac (at least partially), the disappearance of 1 single acetylation group remains ambiguous for its location. Whereas, it is clear that the naa40 KO appears to be the only construction highlighting the disappearance of one acetylation group, this unique result is not sufficient to prove that the acetyl group was on the protein N-terminus. This is also the case for the others constructions where clear MS/MS spectra must be provided for each construction in order to confirm the authors results and the exact acetylation group (or absence) location. This is also the case for figure 5 page 8.

Page 10, lines 358-363, the authors conclude to the N-myristoylation of the Naa40(L) but the glycine could also be a substrates for N-terminal acetylation typical NatA substrate). Since this protein could be subject to few different N-terminal modifications, it would be interesting to also investigate the present (or absence) of the N-terminally acetylated form and/or the non-modified N-terminus. Was it characterized and what is the proportion of each expected proteoforms (related to the three different possible N-terminal modifications)?

Page 11, line 389, the related experiment (using doxycycline) should be at least available in the supplementary Mat&Met.

In the discussion (Page 15, lines 464-467), the authors mention that the shorter Naa40(S) is not expected for different species including A. thaliana. Checking at different publicly available databases such as PPDB (ppdb.tc.cornell.edu) or N-terdb (n-terdb.i2bc.paris-saclay.fr), this conclusion does not appear to be that clear. Indeed, PPDB highlights 2 characterized peptides for the AtNaa40p (AT1G18335) of which one is starting at position 28 with an N-terminus sequence of MDLKKRR. The same peptide is also available in N-terdb. These two distinct sources suggest also the presence of these 2 proteoforms in A. thaliana. Then, the discussion linked to the AtNaa40p (at least lines 461-467 and 477-4836) should be modified and updates considering the experimental results previously mentioned.

Check the sentence starting page15 line505.

Reviewer 4 Report

The paper by Jonckheere & Van Damme describes a detailed study of the functions and localization of two proteoforms of Naa40, conserved Nα-acetyltransferase. The paper provides the reader with interesting insights into the function and regulation of these slightly different forms of the enzyme and in general, the experiments seems to be carefully done, the results convincing, and the conclusions are well founded.  

My major criticisms to this paper come with the writing style. In my opinion most sentences are unnecessarily long, which is again the common criteria for scientific writing, are often twisted and, in some cases, also ambiguous. This is a problem found mostly in the abstract and the Introduction section. I’ll provide the authors with a few examples.

The first sentence in the abstract reads “NatD consist of the evolutionary conserved N-alpha acetyltransferase catalytic subunit Naa40p and is among the most selective N-terminal acetyltransferase (NAT) complexes identified to date”. This sentence conveys the impression that NatD is also a complex. However, in the introduction it is said “However, N-alpha acetyltransferase 40 (Naa40p) also known as NatD, Nat11, Patt1 or Nat4, was shown to be active as a 56 monomeric enzyme.”. Also, it would be nice to define at the very beginning the size of the long form and, from this, define the short one.

The paragraph in lines 47-51 (4.5 lines, not a single period) says essentially nothing.

There is an abuse of unnecessary brackets, which complicates reading. A few examples: “ was recently found to partially account for the observed variations in (the extent of) Nt-acetylation patterns”; “more recent themes about (partial) substrate redundancies...”; studying the (alternative) translation initiation landscape of mice and men..”; and many others…

Yeast nomenclature needs careful revision. NAA40 (not naa40) is an alias for NAT4, the actual accepted gene name. This should be clear in the introduction.  In general, the genetic yeast nomenclature is not respected. I suggest the authors to have a look at the classical paper from Mike Cherry (Cherry JM. Genetic nomenclature guide. Saccharomyces cerevisiae. Trends Genet. 1995 Mar:11-2. PMID: 7660459). Wild type (dominant) alleles are denoted in capital letters. Thus, ynna40 cannot be used to refer to the chromosomal native gene. Similarly, ynaa40-Δ should be "yeast naa40Δ" (line 91). The sentence “pBEVY-U-hNAA40 [20] and pBEVY-U-hNAA40S expression vectors were transformed in the ynaa40S. cerevisiae (Y16202) MAT alpha strain (Euroscarf) (ynaa40-Δ), to generate..” (lines 682-684) has at least two problems. 1) “MAT” is a genotype (MAT locus) and must be in italics. 2) Expressions vectors DO NOT transform cells. Expression vectors are introduced into cells that, thereby, become transformed. By the way: genes do not have substrates (lines 117-118). The encoded proteins have.

Finally, I believe the title could be improved: it says little about the major contributions of this paper.

I encourage the authors to improve these formal aspects that detract from the intrinsic remarkable quality of the work.

Round 2

Reviewer 2 Report

The author has addressed my concerns and I feel the paper is now ready for publication.

Author Response

We are happy that all concerns raised by reviewer 2 were appropriately addressed and acknowledge that the review process has improved the quality of the work. We would also like to express our appreciation for the time spend by the reviewer to review this work.

Reviewer 3 Report

First, I would like to thank the authors for the updates made to their manuscripts.

Considering my comments about “similar results (for mice) [that] have been previously observed and detailed in a dedicated article”, I probably cut too short my thought about the “real novelty” of their Naa40p dual localisation. Indeed, my comments were driven by the article published in 2007 by Chun et al. (doi: 10.1016/j.bbrc.2006.11.131) that described two splicing variants of the Naa10p (indeed, not Naa40p) that localise both at the nucleus and the cytoplasm. That may just be good to mention this previous study and similarity…

Considering the MS/MS spectrum displayed, I remain convinced that both the AAMDAVCAK and the RLEERAAMDAVCA spectra must be provided in the main manuscript for clarity.

Except these 2 points, the corrections provided by the authors are relevant and correspond to my expectations.

Author Response

Response to Reviewer 3 comments – revision 2

First, I would like to thank the authors for the updates made to their manuscripts.

Considering my comments about “similar results (for mice) [that] have been previously observed and detailed in a dedicated article”, I probably cut too short my thought about the “real novelty” of their Naa40p dual localisation. Indeed, my comments were driven by the article published in 2007 by Chun et al. (doi: 10.1016/j.bbrc.2006.11.131) that described two splicing variants of the Naa10p (indeed, not Naa40p) that localise both at the nucleus and the cytoplasm. That may just be good to mention this previous study and similarity…

We are happy that the reviewer appreciated the revisions made to improve the manuscript and also thank the reviewer for further clarifying his/her previous comment. We agree that the analogy with the two Naa10p proteoforms is indeed intriguing and now mention this finding in our discussion (see below text) and additionally reference the study of which note as well as another study reporting on differentially localized isoforms of hNaa30p.

“Interestingly and in analogy, besides the previous report on differentially localized hNaa30p isoforms raised upon alternative splicing [1], a differential subcellular localization and stability was previously also reported in the specific case of mouse Naa10p isoforms differing at their C-terminus and with mNaa10pS being an alternative splicing isoform specific to mouse found to be differentially regulated [2].”

Considering the MS/MS spectrum displayed, I remain convinced that both the AAMDAVCAK and the RLEERAAMDAVCA spectra must be provided in the main manuscript for clarity.

As suggested by the reviewer, we now provide the MS/MS spectrum of the peptide indicative of hNaa40pL expression as a supplementary figure (i.e. Figure S1).

Except these 2 points, the corrections provided by the authors are relevant and correspond to my expectations. The author has addressed my concerns and I feel the paper is now ready for publication.

We would like to thank the reviewer for his/her time spend reviewing this work and we acknowledge that the review process has improved the quality of the work.

References

  1. Varland, S. et al. (2018) Identification of an alternatively spliced nuclear isoform of human N-terminal acetyltransferase Naa30. Gene 644, 27-37.
  2. Chun, K.H. et al. (2007) Differential regulation of splicing, localization and stability of mammalian ARD1235 and ARD1225 isoforms. Biochem Biophys Res Commun 353 (1), 18-25.